# New Insights into the Sex Chromosome Evolution of the Common Barker Frog Species Complex (Anura, Leptodactylidae) Inferred from Its Satellite DNA Content

**DOI:** 10.3390/biom15060876

**Published:** 2025-06-16

**Authors:** Lucas H. B. Souza, Juan M. Ferro, Helena M. Milanez, Célio F. B. Haddad, Luciana B. Lourenço

**Affiliations:** 1Laboratório de Estudos Cromossômicos (LabEsC), Departamento de Biologia Estrutural e Funcional, Instituto de Biologia, Universidade Estadual de Campinas, Campinas 13083-863, São Paulo, Brazil; lucashenriquebs18@gmail.com (L.H.B.S.); helena.mattiazzo@gmail.com (H.M.M.); 2Laboratorio de Genética Evolutiva “Dr. Claudio J. Bidau”, Instituto de Biología Subtropical (CONICET-UNaM), Facultad de Ciencias Exactas, Químicas y Naturales, Universidad Nacional de Misiones, Posadas 3300, Misiones, Argentina; ferrojm@gmail.com; 3Departamento de Biodiversidade, Centro de Aquicultura (CAUNESP), and CBioClima, Instituto de Biociências, Universidade Estadual Paulista, Rio Claro 13506-900, São Paulo, Brazil; haddad1000@gmail.com

**Keywords:** satellitome, chromosomal rearrangements, nucleolar organizer region, repetitive elements, chromosomal homologies

## Abstract

Satellite DNAs (satDNAs) play a crucial role in understanding chromosomal evolution and the differentiation of sex chromosomes across diverse taxa, particularly when high karyotypic diversity occurs. The *Physalaemus cuvieri–Physalaemus ephippifer* species complex comprises at least seven divergent lineages, each exhibiting specific karyotypic signatures. The group composed of *Ph. ephippifer*, Lineage 1B of ‘*Ph. cuvieri*’ (L1B), and a lineage resulting from their secondary contact is especially intriguing due to varying degrees of sex chromosome heteromorphism. In this study, we characterized the satellitome of *Ph. ephippifer* in order to identify novel satDNAs that may provide insights into chromosomal evolution, particularly concerning sex chromosomes. We identified 62 satDNAs in *Ph. ephippifer*, collectively accounting for approximately 10% of the genome. Notably, nine satDNA families were shared with species from distantly related clades, raising questions about their potential roles in anurans genomes. Among the seven satDNAs mapped via fluorescent in situ hybridization, PepSat3 emerged as a strong candidate for the centromeric sequence in this group. Additionally, PepSat11 and PepSat24 provided evidence supporting a translocation involving both arms of the W chromosome in *Ph. ephippifer*. Furthermore, a syntenic block composed of PepSat3, PcP190, and PepSat11 suggested an inversion event during the divergence of *Ph. ephippifer* and L1B. The variation in signal patterns of satDNAs associated with nucleolar organizer regions (NORs) highlights the complexity of NOR evolution in this species complex, which exhibits substantial diversity in this genomic region. Additionally, our findings for PepSat30-350 emphasize the importance of validating the sex-biased abundance of satDNAs.

## 1. Introduction

Repetitive DNAs are crucial components of eukaryotic genomes and can be categorized into transposable elements (TEs), which are typically dispersed throughout the genome; satellite DNAs (satDNAs), which are primarily organized in tandem repetitive DNA, and multigene families, such as ribosomal DNA (rDNA) and histone gene families [1,2,3,4]. Commonly, satDNAs are subject to lower selective pressures compared to other sequence types, and they exhibit a higher divergence, leading to a wide diversity of sequences with varying abundances [5]. On the other hand, sequences of the same satDNA family tend to be homogeneous due to their concerted evolution [4,5,6,7]. Hence, related species may exhibit significant variation in their satDNA profiles [2,8,9], making it intriguing to study the evolution of these sequences and their potential roles in clade divergence and speciation processes.

Recent studies on the satellitomes of various species that provided empirical data contributed to analyses of chromosomal evolution. Significant advances, for instance, have been made in B chromosome differentiation [10,11,12], in the identification of previously underestimated sex chromosome heteromorphism [13,14,15,16], and in the study of hybrid zones and speciation processes [17,18]. In this context, we explored the satellitome of a group of Neotropical frogs that exhibits a complex evolutionary history, involving sex chromosome differentiation and an ancient secondary contact between divergent lineages: the *Physalaemus cuvieri*–*Physalaemus ephippifer* species complex.

This species complex comprises at least seven well-supported genetic lineages [19,20,21], with one of its most remarkable features being its karyotype diversity. Despite having a conserved diploid chromosome number (2n = 22), this taxonomic group exhibits significant variation in the number and/or distribution of nucleolar organizer region (NOR) [20,22,23] and chromosomal clusters of 5S rDNA [22,24,25], PcP190 satDNA [18,25,26], PepBS satDNA [21,27], and the U2 small nuclear RNA (snRNA) gene [28]. Therefore, the study of repetitive elements has proven to be valuable for the comparative analysis of the evolutionary lineages within this group.

*Physalaemus ephippifer* represents one of the genetic lineages from this species complex and is particularly interesting for presenting a heteromorphic ZW sex chromosome system [22,26,27], a rare condition among karyotyped anuran species; for reviews, see [29,30,31,32]. Recent studies have revealed evidence of secondary contact between *Ph. ephippifer* and a lineage lacking differentiated sex chromosomes, informally referred to as Lineage 1B of ‘*Ph. cuvieri*’ (L1B), which also belongs to the *Ph. Cuvieri*–*Ph. ephippifer* species complex. This contact was hypothesized to have given rise to a new system of sex chromosomes, which has been identified in specimens currently found in a region designated as SPAB-VNM-TS/Imp-DE, hereafter referred to as CZ Pep-L1B. Both *Ph. ephippifer* and CZ Pep-L1B specimens exhibit a ZZ/ZW sex chromosome system; however, the Z and W chromosomes of *Ph. ephippifer* differ from those found in the CZ Pep-L1B [18,21].

The heteromorphism of the Z and W chromosomes of *Ph. ephippifer* is related to a terminal segment on the short arm of the W chromosome (Wp), which exhibits a NOR adjacent to a heterochromatic block and is absent on the Z chromosome [22]. Furthermore, an evident cluster of the PcP190 satDNA is distributed on the long arm of the W chromosome (Wq), but is not found on the Z chromosome [26]. On other hand, the Z and W chromosomes of CZ Pep-L1B are very similar to their homologous chromosome 9 of L1B and are less heteromorphic than those of *Ph. ephippifer*, with the most notable difference being a NOR exclusively found on Wp [21]. Based on the chromosome markers available to date, Souza et al. [21] hypothesized that the W chromosome of the CZ Pep-L1B originated from the recombination of chromosome 9 of L1B and the W chromosome of *Ph. ephippifer*. A detailed characterization of these chromosomes, however, is still necessary to evaluate this hypothesis. Since first-generation hybrids between *Ph. ephippifer* and L1B have not been found in the already-sampled localities and the differences in sex chromosome systems may have played an important role in species diversification, studying and characterizing new chromosome markers is particularly interesting in this group. Therefore, in our present work, we characterized the satellitome of *Ph. ephippifer* with the aim of identifying new satDNA families, enabling further comparative analyses.

## 2. Materials and Methods

### 2.1. Specimens

We sequenced the genomes of one male and one female of *Ph. ephippifer* (ZUEC 24908/SMRP 252.156 and CFBH 46750/SMRP 252.166, respectively) and mapped by fluorescent in situ hybridization (FISH) the satDNAs identified in the genomic analysis to chromosomes of representatives of *Ph. ephippifer*, L1B, and the CZ Pep-L1B (refer to Table 1 for the collection accession numbers of the samples). Most tissue samples and chromosome preparations used in this study were obtained from previous works and are available in the tissue and cell suspension collection ‘Shirlei Maria Recco-Pimentel’ (SMRP), housed at the Laboratory of Chromosomal Studies (LabEsC), University of Campinas, São Paulo, Brazil. One specimen from Riachão, Maranhão state, Brazil (CFBH 46777/SMRP 92.421; Table 1), was collected for this work under a permit issued by Chico Mendes Institute for Biodiversity Conservation/Biodiversity and Information System (ICMBio/SISBIO; permit number 32483). Chromosome preparations were obtained from this specimen under authorization of the Committee for Ethics in Animal Use of the University of Campinas (CEUA/UNICAMP; permit number 5723-1/2021). Briefly, we injected the collected specimen intraperitoneally with 2% colchicine (0.02 mL/g body weight), and after 4 h, euthanized it via cutaneous administration of 2% lidocaine (an overdose of 50 mg/g body weight). The intestine was then removed to obtain chromosome preparations following the protocol described by King and Rofe [33], with the modifications described in Gatto et al. [34].

### 2.2. Whole Genome Sequencing and Primary Analysis

Genomic DNA (gDNA) was extracted from the liver of one male and one female of *Ph. ephippifer* following the standard phenol-chloroform method. Genome sequencing was performed by Novogene (Hong Kong, China) using the Illumina NovaSeq 6000 platform, yielding 294,125,536 and 450,288,736 raw reads (150 bp, paired-end) from the female and male samples, respectively. Low-quality sequences (Q < 20) were filtered using Trimmomatic v0.39 [33] with the TruSeq3 adapter option for paired-end reads, and the remaining parameters were set to default on the Galaxy server [34]. The raw reads are available in the Sequence Read Archive (SRA) of GenBank under the accession numbers (SRR33457058, SRR33387195, and SRR33387196). The genome sizes of male and female *Ph. ephippifer* were estimated using k-mer analysis. Briefly, canonical k-mers (k = 21) were counted and exported as histograms with Meryl [35], and the resulting data were used as input for GenomeScope2 [36]. All the analyses were conducted on the European Galaxy server (https://usegalaxy.eu; acessed on 6 October 2023) [37].

### 2.3. RepeatExplorer Analysis

To optimize our selection of satDNAs in the RepeatExplorer analysis, we tested different parameter settings. In all cases, RepeatExplorer was consistently able to process fewer than 3,000,000 random reads from our samples. Therefore, we provided a total of 6 million paired-end reads from the male and female libraries (3 million pairs of reads per sample)—exceeding the number of reads the program could analyze—to ensure the maximum input capacity was utilized. The reads were preprocessed following the protocol described by Novák et al. [38] (https://repeatexplorer-elixir.cerit-sc.cz/; acessed on 6 October 2023). Subsequently, we ran RepeatExplorer2 [38] with default parameters to identify repetitive elements. The analysis was performed in three rounds: the first using only male sequences, the second using only female sequences, and the third as a comparative analysis combining sequences from both samples.

We constructed a library containing the consensus sequences of all candidate satDNAs inferred by RepeatExplorer, including both high- and low-confidence candidates. Clusters lacking dense or ring-shaped graphs, characteristic of tandem repetitive elements [39,40], were excluded. To identify repetitive genes, such as ribosomal RNA and transfer RNA genes, as well as Tes, we utilized BLASTn, BLASTx (available at https://blast.ncbi.nlm.nih.gov/Blast.cgi; acessed on 2 February 2024), Dfam (available at https://dfam.org/home; acessed on 2 February 2024), and tRNAscan-SE v2.0 [41,42]. These sequences were excluded from the library. The remaining candidate satDNA sequences were dimerized and compiled into a FASTA file (custom library). For elements with monomers shorter than 100 bp, additional copies were added until the total sequence length exceeded 200 bp.

### 2.4. RepeatMasker Analysis

We used the RepeatMasker program (https://www.repeatmasker.org/; acessed on 20 January 2025) [43] to characterize the sequences included in the custom library generated from the RepeatExplorer analysis. For this purpose, we used the male and female reads (3 million pairs of reads each) combined into a single file as input. The analyses were performed with default parameters and using RMBlast as the sequence search engine. The generated outputs were used as an input for ParseRM.pl (available on https://github.com/4ureliek/Parsing-RepeatMasker-Outputs; acessed on 23 January 2025) to enhance the visualization of the RepeatMasker analysis.

Additionally, we ran RepeatMasker separately using male and female reads to assess potential differences in the abundance and divergence of satDNA families between the sexes. We calculated the sex ratio of abundance for each satDNA by dividing the female abundance by the male abundance. We expected that satDNA families with higher values in the female (at least 2-fold abundance) would be associated with the W chromosome, while those with lower abundance in the female (ratio less than 0.75) would be candidates for association with the Z chromosome.

### 2.5. Criteria Used for satDNA Designation and Variants Delimitation

The denomination of the new satDNA families was developed following the criteria proposed by Ruiz-Ruano et al. [44], with some modifications. SatDNAs were named using the genus initial and the first two letters of the specific epithet (***P**hysalaemus **ep**hippifer*), followed by “Sat”, a number based on decreasing abundance from the RepeatMasker output (combined male and female files), a hyphen, and a number indicating the length of the most abundant consensus sequence, for example, PepSat1-21.

The custom library was further annotated with BLASTn searches in NCBI. We preserved the nomenclature of the satDNA families previously identified in other studies, adding the term “v-Pep” to the original name. In these cases, we compared the available sequences with the repeats from *Ph. ephippifer* using BioEdit Sequence Alignment Editor v7.2 [45] to estimate pairwise similarity.

For some clusters, the RepeatExplorer output revealed the high variability of sequence composition (i.e., the presence of variants within the same satDNA family), which could be inferred by lower k-mer scores [38]. We examined the individual contigs of these satDNA clusters (>0.4) to identify sequence differences using BioEdit. If the identified distinct sequences showed 80–94% similarity, we considered them variants within the same family. Subsequently, we used the dimerized consensus sequences of the different variants in RepeatProfiler [46] with default parameters to validate and analyze their distribution between the sexes. Variant repeats were labeled with the suffix “v-” followed by a letter to distinguish these sequences, starting with “a” (e.g., PepSat5-1310v-a and PepSat5-1310v-b).

Finally, we searched for the presence of the repetitive sequence PepBS (MW314578; MW314597), previously described by Gatto et al. [27] in *Ph. ephippifer*. Each cluster identified as a candidate satDNA by the RepeatExplorer analysis was examined for the presence of PepBS monomers using the RepeatMasker custom search2 tool, integrated into the RepeatExplorer platform.

### 2.6. Chromosome Mapping of satDNAs by Fluorescent In Situ Hybridization

Based on the consensus sequence of seven candidate satDNAs (refer to the nucleotide sequences in GenBank using the accession numbers PV463926, PV463927, PV463934, PV463939, PV463947, PV463948, and PV463953), we designed primers in opposite directions using dimerized sequences as input in the Primer3 v4.1.0 software (https://primer3.ut.ee/; acessed on 24 October 2023) [47,48]. The primers were used in PCR assays (Appendix A) to isolate and amplify the target sequences from gDNA extracted from liver tissues of *Ph. ephippifer* using TNES solution (50 mM Tris pH 7.5, 400 mM NaCl, 20 mM EDTA, 0.5% SDS, proteinaseK to a final concentration of 100 μg/mL) and precipitated with isopropyl alcohol. PCR-amplified products were purified using the Wizard SV Gel and PCR Clean-Up System kit (Promega) and subsequently sequenced to confirm their identity to the target sequences.

We labeled the purified amplicons with dUTP-digoxigenin or dUTP-biotin via PCR using the PCR Dig Probe Synthesis kit (Roche) or a nucleotide mix containing dUTP-biotin:dTTP in a 1:3 ratio. The labeled probes were precipitated and resuspended in a hybridization solution (50% formamide, 10% dextran sulfate in 2x Saline–Sodium Citrate). For the FISH assays, we used intestine cell suspensions from specimens of *Ph. ephippifer*, L1B, and the CZ Pep-L1B (Table 1), which were available in the SMRP collection from previous works [22,23,25,27]. These samples were dropped onto slides and then the probes were hybridized to the chromosomes following the protocol described by Viegas-Péquignot [49], with a minor modification for the detection of digoxigenin-labeled probes, which was performed using anti-digoxigenin (0.06 µg/mL; Roche). All chromosomes were counterstained with 4′-6 diamidine-2-phenylindole (DAPI; 0.5 µg/mL) diluted in Vectashield (Vector).

Furthermore, for comparative analysis of the chromosomal markers, some chromosome preparations were silver-stained via the Ag-NOR method [50] after FISH. This involved removing the coverslip under running water, applying gelatin and a 50% silver nitrate solution, and incubating at 60 °C for approximately 10 min. We also mapped by FISH the PcP190 satDNA [25] and PepBS satDNA [27] using the same protocol described above. In these cases, we used cloned fragments made available in the SMRP collection by the cited authors. Plasmid DNA was obtained from the clones following the protocol described by Sambrook et al. [51], and the fragments of interest were PCR-amplified and labeled through the same protocols above mentioned, using the universal primers T7/SP6 for the PcP190 sequence and the primers PepBS-F/PepBS-R for the PepBS sequence [27]. Finally, all karyotypes were assembled based on the chromosome size and morphology and the distribution of C-bands, NOR, and chromosome clusters of PcP190 and PepBS satDNAs, following the arrangement proposed in previous studies [20,21,22,23,25]. As most specimens used in the present work had been previously karyotyped, the accurate identification of each chromosome was possible. It is worth noting that the inference of homology between the sex chromosomes of *Ph. ephippifer* and chromosome 9 of L1B was originally proposed by Gatto et al. [27], based on chromosome mapping using a probe constructed from microdissected Z chromosomes of *Ph. ephippifer*. To facilitate comparative analysis of the sex chromosomes of *Ph. ephippifer*, previously identified as chromosome pair 8, e.g., [21,22,27], and their homologous chromosomes in L1B and CZ Pep-L1B, which were designated as chromosome pair 9 in earlier studies, e.g., [18,21,23], we renumbered the sex chromosomes of *Ph. ephippifer* as chromosome pair 9.

## 3. Results

In this study, we employed the short-read sequencing of *Ph. ephippifer* to characterize the satellitome of this anuran species. Selected satDNAs were used for chromosomal mapping to improve the resolution of previously proposed hypotheses on chromosomal evolution and to provide new insights into the organization of repetitive elements. Additionally, genome size estimation was performed using GenomeScope2 for both male and female individuals of *Ph. ephippifer*, revealing genome sizes of 2.41 Gb and 2.43 Gb, respectively (Appendix A). Below, we present a detailed description of the identified satDNAs and the results of the chromosomal mapping analyses.

### 3.1. Characterization of the Ph. ephippifer Satellitome

Using the libraries employed in this study and applying the default threshold of 0.01% in RepeatExplorer2, we identified 62 candidate satDNAs in our analysis (Table 2), whose relative abundance ranged from 1.55% (PepSat1-21) to 0.004% (PepSat62-56). Approximately 10% of the analyzed reads comprised satDNA clusters, with nearly half of the satDNA abundance corresponding to five satDNA families (PepSat1–PepSat5; Table 2). The size of monomers ranged from 16 bp (PepSat44-16) to 1546 bp (PepSat6-1546), with a prevalence of monomers shorter than 100 bp (41 of 62; approximately 66%). AT-rich consensus sequences were more frequent than GC-rich sequences (43 of 62; 69.3%), with a high variation in the AT content, which varied from 35.71% (PepSat62-56) to 66.67% (PepSat 45-33), with an average value of 55 (Table 2). The nucleotide divergence for the identified satDNA families ranged from 1.58% (PepSat3-152) to 17.57% (PepSat34-57), with the majority (47 of 62; 75.8%) showing variation of less than 10% (Table 2). Variant repeat units were recognized in nine families of satDNA, which are discussed and described in Appendix A.

Among the 62 identified satDNA families, 7 exhibited a female-biased abundance, with PepSat30-350 showing the highest female:male abundance ratio (2.38; overall abundance of 0.075%). In contrast, six satDNAs displayed a male-biased abundance, with female/male ratios ranging from 0.73 in PepSat11-237 (overall abundance of 0.242%) to 0.03 in PepSat62-56 (overall abundance of 0.004%) (Appendix A). The most abundant male- and female-biased satDNAs were PepSat11-237 and PepSat16-147 (overall abundance of 0.192%), respectively (Appendix A). Notably, except for PepSat11-237, all male-biased satDNAs (PepSat50, PepSat51, PepSat56, PepSat60, and PepSat62) were among the least abundant in the genome (Appendix A). To investigate their potential association with sex chromosomes, some of these satDNAs were chromosomally mapped in FISH assays, as described in the following section.

We identified the correspondence of nine *Ph. ephippifer* satDNAs and previously described families from *Proceratophrys boiei* forming part of PboSat families (Table 2) [50,51]. Among them, two had been originally described as PcP190 in *Ph. cuvieri* [24] and BBR86 in *Anaxyrus boreas* (Baird & Girard, 1852) (referred to as *Bufo boreas*) [52].

In *Ph. ephippifer*, the monomer length of the satDNA variants shared among different species ranged from 30 bp (PboSat22-90-v-Pep) to 190 bp (PcP190), with only the PcP190 monomer exceeding 100 bp (see Table 2). On the other hand, among these nine satDNA families, the most abundant in *Ph. ephippifer* was PboSat8-92-v-Pep (95 bp in *Ph. ephippifer*), which was ranked as the tenth most abundant satDNA in our analysis (Table 2).

Interestingly, the v-Pep repeat unit regarding the PboSat12-91 was nearly half the size (i.e., 45 bp) of the repeat unit from *Pr. boiei* (i.e., 91 bp). By dividing PboSat12-91 into two fragments of similar sizes, we identified a region that was highly similar (93%) to the 45 bp monomer of v-Pep, whereas the remaining segment exhibited only 68% similarity (Appendix A). Additionally, BBR86-v-Pep monomers were 88% similar to those found in *A. boreas* and 97% similar to the monomer of the PboSat14-41.

### 3.2. Chromosome Mapping of Satellite DNAs in the Clade Ph. ephippifer–CZ Pep-L1B–L1B of ‘Ph. cuvieri’

Of the 62 satDNAs identified by RepeatExplorer in *Ph. ephippifer*, 7 were mapped for the first time using FISH. To aid in the comparative analysis, we renumbered the chromosomes in the *Ph. ephippifer* karyotype to reflect the hypothesis of chromosomal homology supported by Gatto et al. [27]. Consequently, the Z and W chromosomes of *Ph. ephippifer*, which were identified as chromosome pair 8 in earlier studies, e.g., [21,22,27], are here considered pair 9.

PepSat3-152 was the most abundant satDNA mapped by FISH, being found on the primary constriction of the centromeric region of all chromosomes of *Ph. ephippifer* (Figure 1A), L1B (Figure 1B), and CZ Pep-L1B (Figure 1C). On other hand, the satDNA PepSat4-149 displayed a non-clustered or dispersed pattern across the entire chromosomal complement of both sexes of *Ph. ephippifer*, being absent on the NORs of the Z and W sex chromosomes (Appendix A).

Interestingly, the PepSat11-237 probe exhibited FISH signals exclusively on the sex chromosomes of *Ph. ephippifer*, which presented distinct numbers of detected sites. The pericentromeric region of the long arm of both the Z and W chromosomes was detected by this probe, while only the Z chromosome showed an additional interstitial region detected on the short arm (Figure 2A). When hybridized with metaphases from specimens of L1B, this probe exclusively detected the pericentromeric region of the short arm of both homologs of chromosome 9 on either males or females (Figure 2B). Similarly, the Z and W chromosomes of representatives from the CZ Pep-L1B lineage showed enrichment of the PepSat11-237 probe on a pericentromeric region (Figure 2C), adjacently to a PcP190-1a cluster previously mapped by Souza et al. [18]. Additionally, variation in the signal intensity of the PepSat11-237 probe was observed across different specimens in all analyzed groups (Appendix A). All sites containing this sequence coincide with the heterochromatic blocks identified in previous studies [21,22,23].

Similarly, the PepSat24-620 probe also revealed distinct patterns on the Z and W chromosomes of *Ph. ephippifer*. On the Z chromosome, signals were observed in the terminal region of the long arm, adjacent to the NOR, and a weaker signal was occasionally detected in the interstitial region flanking the NOR. In contrast, on the W chromosome, a cluster enriched with the PepSat24-620 sequence was identified adjacent to the NOR, but located on the short arm (Figure 3A), and it coincided with a C-band previously described by Nascimento et al. [23]. In this case, we identified this arm as the short arm of chromosome W through double-FISH assays with the PepSat11-237 and PepSat24-620 probes, which revealed that their signals are located on opposite arms (Inset in Figure 3A). In L1B, a single terminal site was observed on the long arm of chromosome 9, proximal to the NORs, both on males and females (Figure 3B). In addition, in specimens from the CZ Pep-L1B lineage, the Z and W chromosomes displayed terminal clusters of the PepSat24-620 on the long arm, adjacent to the NORs, similarly to the other groups (Figure 3C). In some metaphases of *Ph. ephippifer*, weak probe signals of this sequence were also visible on the short arms of chromosomes 6 and 10, near the telomere and centromere, respectively.

Clusters of both PepSat16-147 (Figure 4A–C) and PepSat25-282 (Figure 4D–F) were associated with NORs in all groups analyzed; however, differences regarding their probe signals in FISH assays were noted. Firstly, the PepSat16-147 probe revealed a cluster not associated with NOR in an interstitial region of the Z chromosome of *Ph. ephippifer*, in addition to the clusters that colocalized with the NORs on the Z and W chromosomes of this species (Figure 4A). Moreover, the signals revealed by this probe varied in both intensity and position relative to the secondary constrictions of the NORs. While the PepSat16-147 probe signals coincided with NORs in most cases, in chromosome 8 of L1B specimens, the cluster of PepSat16-147 flanked the NOR secondary constriction (inset in Figure 4B). In the CZ Pep-L1B specimens, clusters enriched with this sequence and associated with the NORs on autosomes 7 and 8 presented weaker probe signals compared to those on the sex chromosomes (Figure 4C). Lastly, in the CZ Pep-L1B karyotype, the NOR located on the short arm of the W chromosome did not exhibit any signals of this satDNA (Figure 4C). In contrast to PepSat16-147, PepSat25-282 was consistently and exclusively associated with all NORs in *Ph. ephippifer*, L1B, and CZ Pep-L1B (Figure 4D–F), and the signals of the PepSat25-282 probe did not show a notable variation of intensity among the NORs and were slightly weaker than those revealed by the PepSat16-147 probe (Figure 4).

Finally, since PepSat30-350 exhibited the greatest difference in abundance between the sexes (for details see Appendix A), we mapped this sequence on the chromosomes of *Ph. ephippifer* specimens. The probe consistently hybridized to the pericentromeric region of the long arm of autosome pair 3 in all analyzed samples, as well as to a polymorphic site on chromosome 5, specifically in the pericentromeric region of its short arm. In the karyotypes of the three females and one of the two males that were subjected to FISH, the PepSat30-350 probe signal was found on only one homologue of chromosome 5. However, in the second studied male, whose genome was sequenced using Illumina, probe signals were observed exclusively on the chromosomes of pair 3, with no detected signals on the chromosomes of pair 5 (Figure 5). Therefore, the chromosome mapping did not validate PepSat30-350 as a sex-linked marker.

### 3.3. Supercluster Formed by rDNA and Satellite DNAs

Upon analyzing the PepBS repetitive element, we identified a contig of a cluster (CL68 in Figure 6) not annotated as a tandem repeat by RepeatExplorer, which was 78% similar to a PepBS sequence from GenBank (accession number MW314597). This cluster is part of a supercluster containing five additional clusters, two of which correspond to PepSat16-147 and PepSat25-282, which were associated with NORs in the FISH assays, as previously mentioned (Figure 4). Two other clusters in this supercluster showed similarity to nucleolar 40S rDNA, as annotated by RepeatExplorer (Figure 6).

## 4. Discussion

### 4.1. Genome Size Estimation

This is the first report of genome size for this species (2.4 Gb), which falls within the range of genome sizes reported for other leiuperines. For example, *Physalaemus biligonigerus* has a genome size of 1.9 pg (1.8 Gb), *Ph. nattereri* has 1.8 pg (1.7 Gb), *Pseudopaludicola mystacalis* has 2.4 pg (2.4 Gb), and *Pleurodema bibroni* has 2.6 pg (2.5 Gb); see Gregory [55] (www.genomesize.com; accessed on 9 May 2025) and references therein. Moreover, *Engystomops pustulosus* has a genome size of 2.2 Gb, based on the NCBI RefSeq assembly GCF_040894005.1.

### 4.2. General Aspects of the Satellitome of Ph. ephippifer

In our analysis, we provide a description of the satellitome of *Ph. ephippifer*, making it one of the few anuran species whose satellitome has been analyzed, alongside *Pr. boiei* [52,53]. A total of 62 satDNAs were identified in *Ph. ephippifer*, of which 53 are described here for the first time. The remaining nine had been previously identified either in the satellitome of *Pr. boiei* [52,53], or in earlier studies of *Ph. cuvieri* [25] and *Bufo boreas* [54], which used different approaches to isolate and recognize repetitive sequences. In our study, five families of satDNAs stand out as the most abundant, following a pattern of predominance where a few families are more abundant than most others. This pattern is commonly reported in the genomes of other species with multiple satDNA families, e.g., [11,52,56].

Some of the satDNA families identified in *Ph. ephippifer* exhibited a high level of divergence (over 10%) among the repeats, and they can be classified into three major categories. The first includes satDNAs with smaller monomers (<50 bp), characterized by well-defined variants with some frequent mutations and several less frequent ones (e.g., PepSat8-36, whose monomers show 10.88% divergence). The second category includes satDNAs with variable regions concentrated within specific portions of the monomer (e.g., PepSat6-1546). The third category encompasses highly variable satDNAs according to RE output, with low k-mer coverage values, for example the PepSat7-902 (TAREAN k-mer coverage = 0.26), which suggest the absence of a predominant monomer. These three patterns have been previously described in studies of other animal groups, and in some cases, the high variability of certain satDNAs may be associated with the rapid amplification of repetitive elements or the divergence of a satDNA from different regions of the genome, hindering their homogenization by concerted evolution, e.g., [57,58].

Another remarkable finding refers to the identification of nine satDNAs that are shared between *Ph. ephippifer* and *Pr. boiei*. *Physalaemus* and *Proceratophrys* belong to different anuran families, whose most recent common ancestor dated from more than 60 million years ago [59], which suggests that the shared satDNAs are at least this old and are likely present in several anuran genera. Similarly, recent studies have shown that the BamH1-800 satDNA family is widely distributed among species of Bufonidae [60], and the PcP190 satDNA is found in different families within Hyloidea (see [24,34,52], and references therein). SatDNAs can exhibit species-specific patterns due to distinct amplification events, as proposed by the library hypothesis [9,61]. Therefore, satDNAs shared between species are valuable for karyotypic comparisons and for inferring chromosomal evolution. In this context, satDNA analyses have been employed in several groups, such as grasshoppers [62] and flies [63].

Curiously, eight out of the nine satDNA families shared by *Ph. ephippifer* and *Pr. boiei* contain monomers of small sizes (30–95 bp). Given the significant phylogenetic distance between these species, this is an intriguing finding, as it is generally expected that the greater the evolutionary divergence between groups, the higher the likelihood that their sequences will accumulate mutations and diverge. This raises the question of whether these diminutive and conserved satDNAs have some biological function, a possibility already noted in other groups with small satDNAs that transcribe siRNAs or perform other structural roles, as reviewed in [3,4,61]. However, further investigation into the roles of these satDNAs is required before drawing more definitive conclusions.

### 4.3. Common Challenges in the Study of Satellite DNA Revealed by the Analysis of Ph. ephippifer

In recent years, advancements in sequencing technologies and bioinformatic tools have made the analysis and development of genomic data more accessible, and numerous studies across various taxonomic groups have successfully analyzed the satellitome of non-model species [15,16,44,64,65,66,67,68,69,70,71]. Regarding anurans, only two studies had applied this approach until now, both focusing on the same species, *Pr. boiei* [52,53], for which no satDNA had been previously described. Here, we investigated the satellitome of another anuran species, *Ph. ephippifer*, of which two repetitive elements were already known. Although this approach overcame the limitations of previous studies in identifying satDNAs, it raised some questions that we believe are common challenges in many current satDNA studies.

One such question concerns the assignment of multiple names to the same satDNA families. For example, the PcP190 satDNA family was first described by Vittorazzi et al. [25], following a naming convention based on the species from which the sequence was isolated (***P****hysalaemus **c**uvieri*), the location of the sample (**P**almeiras, Bahia state), and the monomer size (**190** bp). Subsequent research on this satDNA family revealed that its monomer contains two distinct regions: a conserved region (CR) of 120 bp and a hypervariable region (HR), which varies greatly in size and composition both intra- and interspecifically. This variability has been used to identify different variants of this family across many Hyloidea species, e.g., [34]. However, recently, two new names were assigned to this satDNA, PboSat3-189 [53] and PboSat02-192 [52], both described for *Pr. boiei*. In our present study, this satDNA could be labeled as PepSat22-190. Under this naming system, new studies on Hyloidea species with the PcP190 sequence will continue to assign different names to the same satDNA, complicating the accurate analysis and comparison of these sequences. Additionally, it is worth noting that a simple listing of satDNAs found in parallel studies, without proper comparative analyses, can also result in more than one description for the same satDNA family (see Appendix A). The study and characterization of repetitive elements has long been challenging, with the issue of multiple nomenclatures being just one of many factors hindering the correct identification of these sequences. Other issues include the formation of chimera sequences, caused by the assembly of distinct sequences together, and the lack of connectivity in larger elements, as highlighted by Šatović-Vukšić and Plohl [72].

Another key point highlighted in our study is the need for caution when determining whether a satDNA is sex-linked. While comparative analyses of male and female satellitomes can help identify satDNAs that are more abundant in one sex, further analyses are required to confirm them as true sex-linked markers. In our study, using FISH, we were able to discard PepSat30-350 as a sex-linked marker, despite observing a significant difference in its abundance between the analyzed female and male genomes (female/male abundance ratio of 2.38). Although the sex-biased abundance may suggest a possible association with the W chromosome, the chromosome mapping of PepSat30-350 onto the karyotype of several specimens revealed that the variation in the abundance of this satDNA in the genomic analysis was due to an additional polymorphic cluster on chromosome 5 in the sequenced female, which was absent in the sequenced male. Another noticeable case concerns the PcP190 satDNA, which is clustered on the sex chromosomes of *P. ephippifer*. This satDNA exhibits a sex-linked difference, characterized by the presence of a prominent cluster revealed by FISH in the pericentromeric region of Wq, while the corresponding region on the Z chromosome is much smaller [26] (present work). Despite being more abundant on the W chromosome, the genomic analysis did not reveal a significantly higher abundance of this satDNA in the female (estimated female/male abundance ratio: 1.07). It is likely that this difference was masked by a notable cluster found on the short arm of chromosome 3, which appears as a strong signal in FISH assays and is present in both sexes [26]. Furthermore, considering that in our analysis more than half of the satDNAs exhibiting sex-biased abundance (7 out of 13) were among the least abundant in the genome, caution is required when associating them with sex chromosomes without proper validation. These values could result from sampling bias, as less abundant sequences are more susceptible to such variations.

In addition to the scenarios discussed above, when searching for sex chromosome markers, it is also important to consider that satDNAs with equal abundance in males and females may be clustered in distinct regions on the sex chromosomes, still serving as informative markers. In the case of PepSat24-620, for instance, no sex-biased abundance ratio was found in the genomic analysis, but this sequence was mapped by FISH to distinct arms of the Z and W chromosomes of *Ph. ephippifer*, highlighting structural heteromorphism between these chromosomes (see further discussion in the next section). In conclusion, we advocate that using the sex-biased abundance ratio could be an interesting starting point for investigating sex chromosome satDNA candidates, but it should not be considered a sufficient approach.

### 4.4. Chromosome Mapping of Satellite DNAs Is Useful for the Comparative Analysis of Physalaemus ephippifer, L1B, and CZ Pep-L1B

The chromosome mapping of satDNAs prospected from genomic analyses revealed distinct distribution patterns, allowing us to identify satDNAs associated with centromeric regions, NORs, and sex chromosomes, and enabling valuable insights into the distribution of repetitive elements within the group comprising *Ph. ephippifer*, L1B, and CZ Pep-L1B, as discussed below.

Additionally, another FISH pattern of satDNAs we found here concerns PepSat4-149, which was not organized into large clusters but rather mapped in a dispersed, pulverized pattern. This characteristic has been increasingly reported following the characterization of many satellitomes, e.g., [44,57,73], and is likely related to the presence of numerous arrays with a low number of tandemly repeated monomers scattered throughout the genome. Ruiz-Ruano et al. [44] suggested that this represents an initial step in satDNA evolution, where the sequence first spreads across the genome, followed by its establishment at new loci and subsequent expansion through mechanisms such as unequal crossing-over, eventually leading to the establishment of new clusters that can be detected in FISH assays.

#### 4.4.1. Centromeric Satellite DNA

Centromeric regions exhibit plasticity and are typically dominated by satDNAs, as reviewed in [9]. In some species, a single satDNA is widely distributed across the centromeres of all chromosomes, e.g., [66,74], whereas in others, different chromosomes may harbor distinct centromeric repeats. In certain cases, transposable elements constitute the main centromeric component [75], and in rare instances, some chromosomes may lack any detectable repetitive elements associated with their centromeres [76]. Notably, the high genomic abundance of a given satDNA does not necessarily imply its centromeric function or localization, highlighting the need for cytogenetic or genomic mapping to confirm its role.

PepSat3-152 represented the satDNA family of *Ph. ephippifer* with the least variation in nucleotide sequence among monomers (mean divergence of 1.58%). When mapped by FISH, this satDNA was found in all the centromeres of the analyzed karyotypes. Considering both its chromosomal location and the high sequence conservation, it is reasonable to infer that this sequence may play a key role in centromere biology in these anurans—a characteristic that has already been identified and thoroughly explored in other groups for various centromeric satDNAs; for reviews, see [3,9,72].

Recently, Da Silva et al. [77] mapped PboSat01-176 in four *Proceratophrys* species, which exhibited (peri)centromeric signals in all chromosomes of the species analyzed. The PboSat01-176 sequence was not found in our analysis of the *Ph. ephippifer* satellitome. But the involvement of additional satDNAs or other classes of repetitive elements in the structural organization of the centromeric/pericentromeric regions of *Ph. ephippifer* cannot be ruled out, warranting further investigation. Given that centromeric organization in anurans remains largely underexplored, additional investigations focused on both PboSat01-176 and PepSat3-152 may yield significant insights.

#### 4.4.2. Satellite DNAs Associated with rDNA

Among the recognized NORs in *Ph. ephippifer*, L1B, and CZ Pep-L1B, some coincide with heterochromatic blocks identified by C-banding (i.e., the NORs on chromosome 9 of L1B, chromosome Z of CZ Pep-L1B, and the long arm of the W chromosome in CZ Pep-L1B), while other NORs do not coincide with C-bands (i.e., the NORs on chromosomes Z and W of *Ph. ephippifer*, on chromosomes 7, 8, and on the short arm of W of CZ Pep-L1B, and on chromosome 8 of L1B) [21,22,23]. In a previous study, Gatto et al. [27] observed that the repetitive DNA PepBS is colocalized with the NORs found in *Ph. ephippifer* as well as with the NORs of L1B. The authors also noted that in L1B, the probe signal was stronger on chromosome 9, which coincides with C-band, than on chromosome 8. In the CZ Pep-L1B specimens analyzed by Souza et al. [21], the NORs on the long arm of the Z and W chromosomes showed a stronger signal of the PepBS probe compared to the other NORs (i.e., NORs on chromosomes 7 and 8).

Here, the satDNAs PepSat16-147 and PepSat25-282 were mapped to the NORs in specimens from the *Ph. Cuvieri*–*Ph. ephippifer* species complex. The supercluster analysis generated by RepeatExplorer, which included 40S rDNA, suggests that these three types of repetitive DNA are part of the intergenic spacer (IGS) of the nucleolar rDNA in *Ph. ephippifer*. Additionally, a cluster containing the PepBS sequence was also found within the same supercluster. It is important to note that these are not the only repetitive elements found in the supercluster, and consequently, not the only ones within this IGS (see Figure 6).

The PepSat25-282 satDNA was associated with all the NORs in *Ph. ephippifer*, L1B, and CZ Pep-L1B specimens, with no notable variation in signal intensity of the hybridized probe. In contrast, the intensity of the PepSat16-147 probe varied across the NORs, as well as the relative positions of clusters of these sequences and the secondary constriction, with L1B representing an important lineage that bears this diversity. In addition, remarkably, this satDNA is absent on the NOR on the short arm of the W chromosome in CZ Pep-L1B, whereas the NOR on the short arm of the W chromosome of *Ph. ephippifer* is highly enriched in this sequence. Therefore, despite both of these NORs not colocalizing with C-bands [21,23], they have differences, which highlights the distinction between the sex chromosomes of *Ph. ephippifer* and the CZ Pep-L1B (Figure 7; Appendix A).

The relationship between nucleolar rDNA IGS and satDNAs has been reported in some other taxonomic groups, including amphibians, butterflies, dipterans, and plants, and the interactions these sequences may facilitate have been discussed [78,79,80,81,82,83]. It has been suggested that the origin of certain satDNAs could be related to specific IGS types, e.g., [80,83], and that satDNAs may play a significant role in the spread of rDNA to different locations in the genome through mechanisms such as the formation of extrachromosomal circular DNAs with subsequent reinsertion or ectopic recombination [5,82,84]. However, although variations in the number, location, and size of NORs have been widely observed among anuran species [85,86,87,88], their origin and association with other repetitive elements remain poorly understood. Given that NORs are among the most relevant features of karyotypic diversity in the *Ph. cuvieri–Ph. ephippifer* species complex [20,21,22,23], it would be promising to investigate the intriguing relationship between nucleolar rDNA and the satDNAs described here in the remaining lineages of the species complex, particularly in L3, which is characterized by a notable NOR polymorphism, or even in Balsas (MA), and has been suggested as a contact zone between L1B and L3 [21,23]. Further comparative analyses of the multiple NORs found in these lineages—ideally incorporating high-resolution cytogenetic techniques such as fiber-FISH—may help clarify the underlying causes of the different levels of NOR variation observed in this group.

#### 4.4.3. Satellite DNAs Associated with Sex Chromosomes

The PepSat11-237 and PepSat24-620 satDNAs are valuable markers associated with the sex chromosomes of *Ph. ephippifer* and their homologous counterparts, providing new insights that allow us to discuss potential rearrangements involved in the differentiation of these chromosomes. Gatto et al. [27] suggested that chromosome 9 of the L1B karyotype is homologous to the Z/W sex chromosomes of *Ph. ephippifer*, based mainly on the mapping of probes obtained from microdissection (referred to as Zqper/8p) [27]. The PepSat11-237 probe revealed the same chromosome sites as were detected by the Zqper/8p probe; however, the signal intensity of PepSat11-237 was much stronger. These results suggest that part of the composition of the Zqper/8p probe may include sequences of the satDNA PepSat11-237.

The satDNA PepSat24-620 was mapped terminally on the long arm of the Z chromosome of *Ph. ephippifer* and on the long arm of chromosome 9 of L1B, adjacent to NORs in both cases. However, in the W chromosome of *Ph. ephippifer*, this probe hybridized on the terminal region of the short arm, also adjacent to a NOR site. This finding suggests the possibility of an intrachromosomal translocation event or a pericentric inversion, in which a segment composed of a NOR and the terminal PepSat24-620 region was transferred from Wq to Wp, a rearrangement that may explain the most prominent difference between the sex chromosomes of *Ph. ephippifer* (event 2 in Figure 7).

The chromosome sites enriched in PepSat11-237, PepSat24-620, and PcP190 comprise a syntenic block, located adjacent to the centromere. While in L1B, this block is found on the short arm of chromosome 9, and the NORs are located in the long arm of this chromosome, in *Ph. ephippifer*, this block is located on the long arm of the Z and W chromosomes, along with the NORs. Such findings corroborate the hypothesis previously raised by Gatto et al. [27] regarding the occurrence of a pericentromeric inversion after the divergence of these clades (Figure 7). However, further studies, including external groups, are still needed to properly assess the origin of this rearrangement.

The chromosome markers PepSat3-152, PepSat11-237, PepSat24-620, and PepSat25-282 mapped in the sex chromosomes of *Ph. ephippifer* and of the CZ Pep-L1B are compatible with the hypothesis proposed by Souza et al. [21] regarding the origin of the W chromosome in the CZ Pep-L1B. Specifically, the W chromosome likely arose from recombination events between chromosome 9 of L1B and the W chromosome of *Ph. ephippifer* [21]. However, the absence of a detectable cluster of PepSat16-147 in the NOR of Wp in CZ Pep-L1B distinguishes this NOR from that of its hypothetical ancestor, the Wq NOR of *Ph. ephippifer*, which exhibits a strong cluster of this satellite DNA (Figure 7). Notably, the Wp NOR of the CZ Pep-L1B karyotype was the only NOR of this karyotype not mapped by the PepSat16-147 probe. Therefore, in the hypothetical scenario proposed for the origin of the W chromosome in CZ Pep-L1B, this NOR would have experienced a significant reduction in copy number of this satDNA sequence after the supposed recombination event. This loss is in line with the expected variability for this type of sequence, as predicted by the library hypothesis [5,61,89].

Regarding the differences between the Z and W chromosomes of *Ph. ephippifer*, previous studies have identified a polymorphism involving the presence of an interstitial C-band in Zp (see [22]) located near a U2 snRNA gene cluster [28] and detected by the Zqper/8p probe in FISH assays [27]. This heterochromatic region is absent in some Z chromosomes and in all analyzed W chromosomes. With the new data collected, we could improve the molecular analysis of this heterochromatic band, as PepSat11-237 and PepSat16-147 clusters were found to coincide with this region.

Therefore, in addition to PcP190, which is more abundant in the W chromosome than in the Z chromosome [26], PepSat11-237, PepSat16-147, PepSat24-620, and PepSat25-282 are also satellite DNAs differentially distributed between the sex chromosomes of *Ph. ephippifer* (Figure 2, Figure 3 and Figure 4; Appendix A). This variation in satDNA composition along the sex chromosomes of *Ph. ephippifer* mirrors patterns that have been commonly observed for heteromorphic sex chromosomes in various organisms, e.g., [13,14,16,90,91].

## 5. Conclusions

Through the satellitome characterization of *Ph. ephippifer*, we have provided new insights into satDNA dynamics in anurans by identifying satDNAs conserved across distantly related species. Additionally, among the satDNAs described, we identified a potential candidate for centromeric satDNA, at least within the *Ph. cuvieri*–*Ph. ephippifer* species complex, and highlighted notable NOR variation associated with distinct repetitive elements in the IGS of 40S rDNA. These findings contribute to a better understanding of the remarkable NOR variation in this species complex. Finally, the chromosomal mapping of satDNAs linked to the sex chromosomes of *Ph. ephippifer*, CZ Pep-L1B, and its homologous pair 9 in L1B suggests the occurrence of chromosomal rearrangements, and supports previous hypotheses of accelerated karyotype evolution within the *Ph. cuvieri*–*Ph. ephippifer* species complex.

## Figures and Tables

**Figure 1 biomolecules-15-00876-f001:**
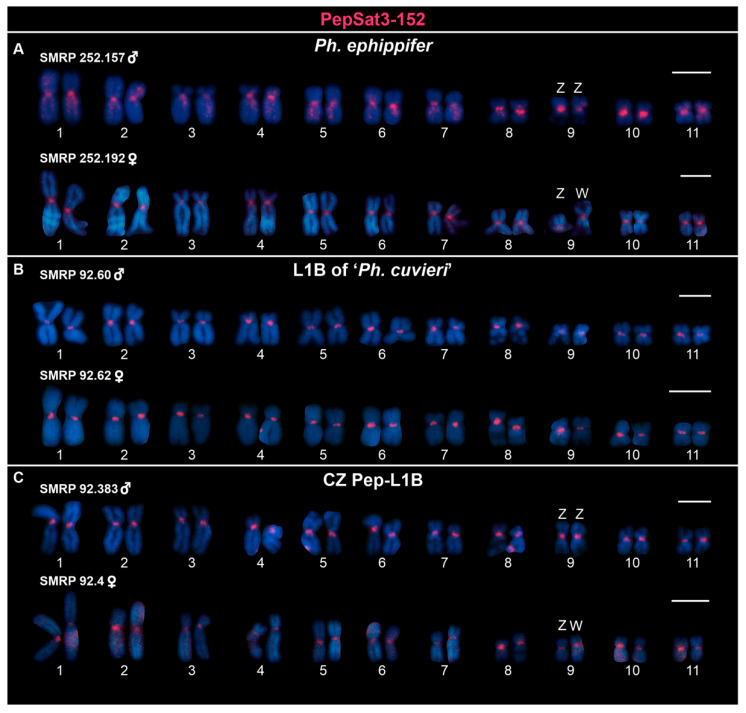
Karyotypes hybridized with the PepSat3-152 probe. (**A**) Specimens of *Physalaemus ephippifer* from Santa Bárbara, Pará, Brazil. (**B**) Samples from Lineage 1B (L1B) of ‘*Ph. cuvieri*’, from Urbano Santos, Maranhão, Brazil. (**C**) Individuals from São Pedro da Água Branca (SPAB), Maranhão, and Dom Eliseu (DE), Pará, both localities being within the known range of the CZ Pep-L1B. Scale bar = 5 μm.

**Figure 2 biomolecules-15-00876-f002:**
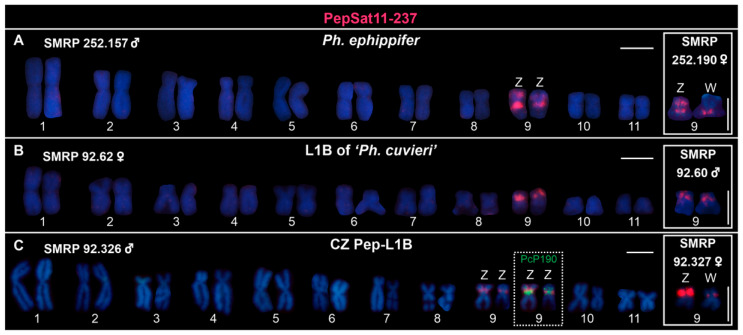
Karyotypes hybridized with the PepSat11-237 satDNA probe. (**A**) Specimen from Santa Bárbara, Pará, Brazil. The inset displays the heteromorphic sex chromosome pair Z and W from a female from the same locality. (**B**) Female from Urbano Santos, Maranhão, Brazil. The inset shows pair 9 from a male from the same site. (**C**) Sample from São Pedro da Água Branca (SPAB), Maranhão. The inset highlights the female Z and W chromosomes marked with the probe. The dashed inset shows the same chromosome pair from the karyogram hybridized with both PcP190 1a and PepSat11-237 probes, where both signals are adjacent, with PcP190 located closer to the centromeric region. Scale bar = 5 μm.

**Figure 3 biomolecules-15-00876-f003:**
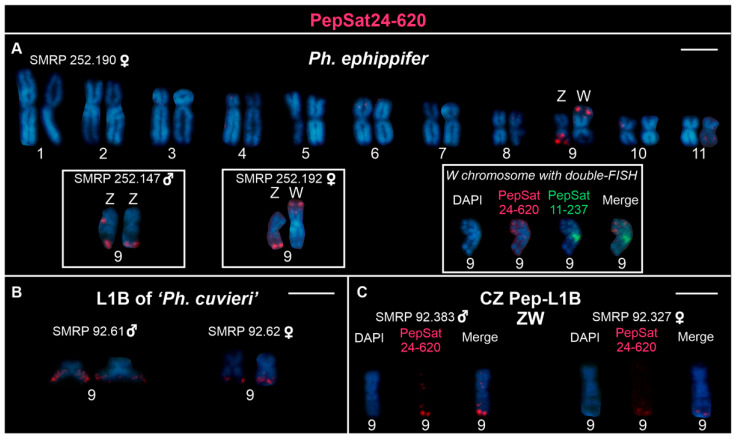
Chromosomes hybridized with the PepSat24-620 probe. (**A**) Female *Physalaemus ephippifer* from Santa Bárbara, Pará, Brazil. The left insets show the sex chromosomes of two additional specimens mapped with the same probe, while the right inset highlights the W chromosome marked with both PepSat11-237 (green) and PepSat24-620 (red). Note that signals are located in opposite arms of the chromosome. (**B**) Chromosome pair 9 of both male and female from Urbano Santos, Maranhão, Brazil. (**C**) Z and W chromosomes of specimens from CZ Pep-L1B. Scale bar = 5 μm.

**Figure 4 biomolecules-15-00876-f004:**
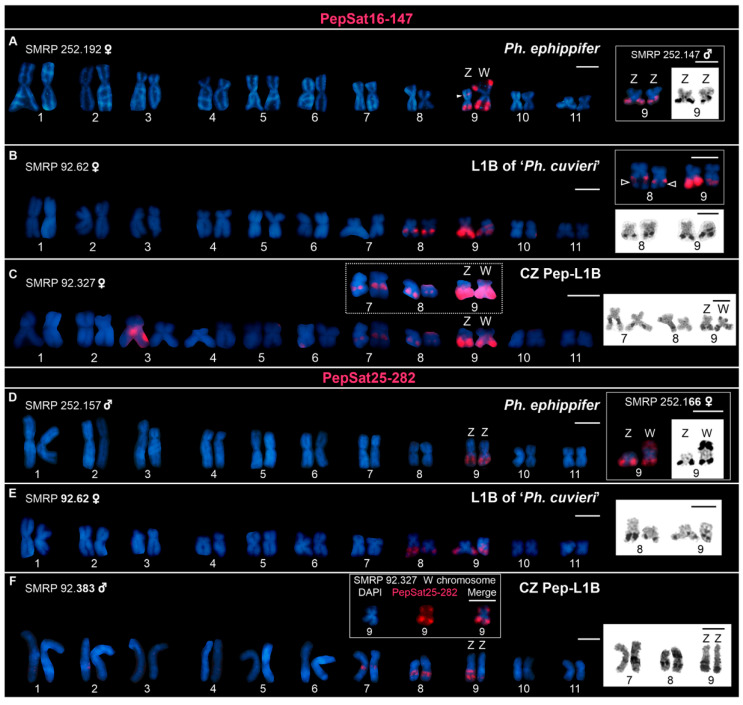
Karyotypes mapped with the PepSat16-147 (**A**–**C**) and PepSat25-282 (**D**–**F**) probes. (**A**) Female *Physalaemus ephippifer* from Santa Bárbara, Pará, Brazil. The filled arrowhead on the Z chromosome indicates the only signal of this probe that does not coincide with NORs. (**B**) Female of Lineage 1B (L1B) of *‘Ph. cuvieri’*, collected in Urbano Santos, Maranhão, Brazil. The top inset shows chromosomes from a less compacted metaphase hybridized with the same probe; note that on chromosome 8, the probe signal does not extend across the entire secondary constriction (empty arrowhead). (**C**) Female metaphase of a specimen of the CZ Pep-L1B lineage, from São Pedro da Água Branca, Maranhão, Brazil. This specimen possesses only one evident NOR-bearing homologue of chromosome 8 (the one on the left in the shown chromosome 8 pairs). Note that the short arm of the W chromosome lacks a cluster of this satellite DNA, although it carries a terminal NOR. The probe signal observed on chromosome 3 is an artifact caused by overlap with the W chromosome in the metaphase spread. The dashed inset exhibits the same chromosome pairs 7, 8, and 9 (ZW) from the karyogram with enhanced brightness for better visualization of the weak signals on chromosome pairs 7 and 8, as well as the difference in intensity compared to the sex chromosomes. (**D**) Male *Ph. ephippifer* from Santa Bárbara, Pará, Brazil. (**E**) Female from Lineage 1B (L1B) of *‘Ph. cuvieri’* from Urbano Santos, Maranhão, Brazil. (**F**) Male specimen from the CZ Pep-L1B, collected in Dom Eliseu, Pará. Insets in (**A**,**C**,**D**) show NOR-bearing chromosomes from other metaphases hybridized with the probes, followed by Ag-NOR staining. The bottom inset in (**B**), as well as the insets in (**E**,**F**), display the same chromosomes from the karyogram after the Ag-NOR assay. Note that the heteromorphisms in NOR size (for example, in pair 8 in (**E**)) and/or number (for example, the ZZ pair in (**F**)) correspond to heteromorphisms revealed by the probes, except for chromosome pair 8 of L1B (shown in (**B**)), in which the PepSat16-147 cluster is adjacent to the NOR constriction (inset in (**B**)). Scale bar = 5 μm.

**Figure 5 biomolecules-15-00876-f005:**
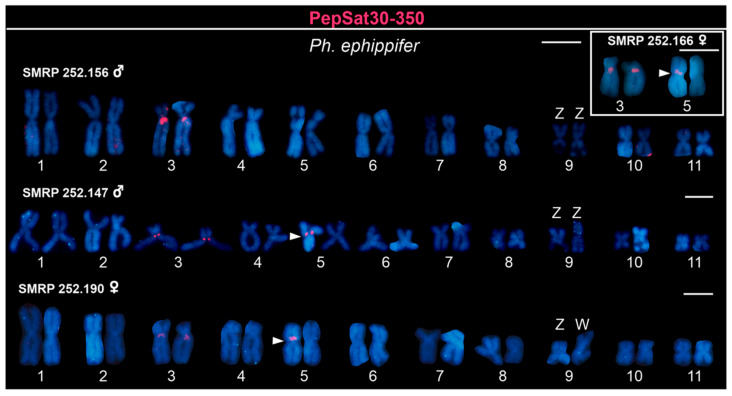
Karyotypes of *Physalaemus ephippifer* from Santa Bárbara, Pará, Brazil, hybridized with the PepSat30-350 probe. The first karyogram and the inset represent the individuals used for Illumina sequencing in this study (i.e., specimens SMRP 252.156 and 252.166). Notably, these individuals exhibit differences in the number of satDNA clusters on chromosome 5, as indicated by the arrowheads. However, upon analyzing a larger sample, we identified this variation as a polymorphism not linked to sex, as the male specimen SMRP 252.147 also exhibits this signal. Scale bar = 5 μm.

**Figure 6 biomolecules-15-00876-f006:**
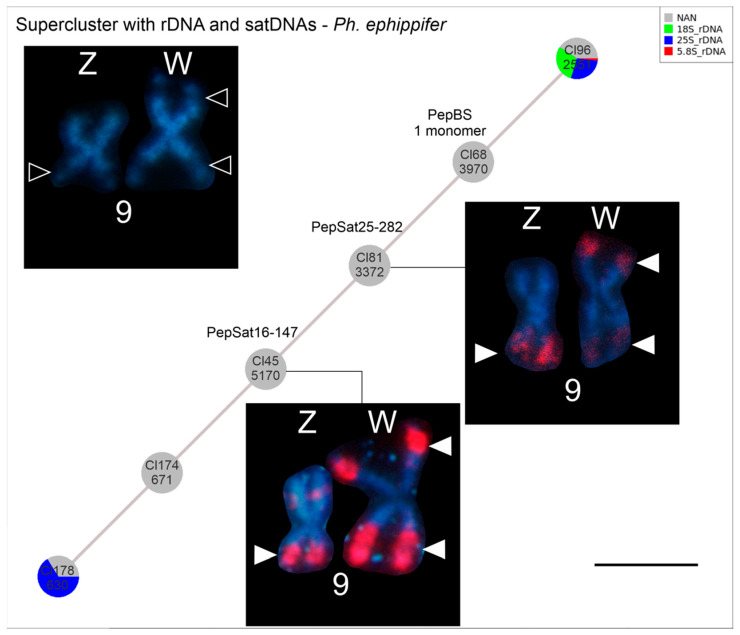
Schematic representation of the *Physalaemus ephippifer* supercluster generated by RepeatExplorer analysis, showing two clusters corresponding to 40S rDNA sequences, connected to three additional clusters previously mapped in *Ph. ephippifer*, namely, PepBS [27], PepSat16-147, and PepSat25-282 (this study). In the top left, the sex chromosomes of *Ph. ephippifer* are shown with DAPI staining, highlighting the secondary constrictions (empty arrowheads). The two other insets in the bottom right depict the chromosomal mappings of PepSat16-147 and PepSat25-282 on the sex chromosomes of *Ph. ephippifer*, coinciding with NOR regions (filled arrowheads). Scale bar = 5 μm.

**Figure 7 biomolecules-15-00876-f007:**
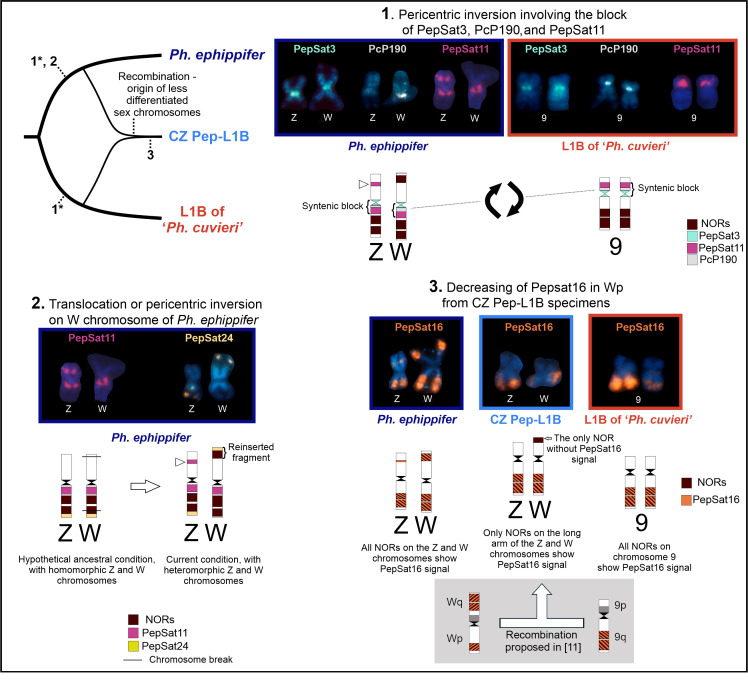
Cladogram representing our hypothesis of chromosomal evolution for chromosome pair 9 of *Physalaemus ephippifer*, Lineage 1B (L1B) of ‘*Ph. cuvieri’*, and CZ Pep-L1B. Ideograms and chromosome images illustrate the main features of these chromosomes, and numbers 1-3 indicate possible evolutionary changes. All FISH images were generated using digoxigenin-labeled probes and rhodamine-conjugated anti-digoxigenin, and artificial colors were assigned to the satDNA probes using Adobe Photoshop 2021. Asterisks (*) indicate that the pericentric inversion involving the syntenic block composed of PepSat3- PcP190-PepSat11 may have occurred either in the lineage of *Ph. ephippifer* or in L1B. Arrowheads indicate a polymorphic PepSat11 cluster of the Z chromosome of *Ph. ephippifer*, which can be absent. The bottom right panel includes a representation of the hypothetical recombination between the W chromosome of *Ph. ephippifer* and chromosome 9 of L1B (gray rectangle), originally proposed by Souza et al. [11], which may have given rise to the W chromosome of the CZ Pep-L1B. Gray blocks indicate Zqper/8p probe signals, as reported in [11]. Note that, according to this hypothesis, the NOR on Wp of CZ Pep-L1B would have originated from Wq of *Ph. ephippifer*.

**Table 1 biomolecules-15-00876-t001:** Locality of origin and accession numbers in the zoological and tissue and cell suspension collections of the specimens analyzed in this work. CZ Pep-L1B: specimens collected in the secondary contact zone between *Ph. ephippifer* and Lineage 1B. PA: Pará state. MA: Maranhão state. ZUEC: Museum of Biological Diversity, University of Campinas (UNICAMP), Campinas, São Paulo state, Brazil. CFBH: Collection ‘Célio F.B. Haddad’, Biodiversity Department, São Paulo State University (UNESP), Rio Claro, São Paulo state, Brazil. HUFMA: Herpetology Collection of the Federal University of Maranhão, Maranhão state, Brazil. MNRJ: National Museum of Rio de Janeiro, Rio de Janeiro state, Brazil. SMRP: Tissue and cell suspension collection ‘Shirlei Maria Recco-Pimentel’, UNICAMP, Campinas, São Paulo state, Brazil. Specimens ZUEC 24908/SMRP 252.156 and CFBH 46750/SMRP 252.166 were used for genome sequencing.

Group	Locality (City-State)	Number of Specimens	Accession Number in Zoological Collection	Accession Number in SMRP Collection
*Physalaemus ephippifer*	Santa Bárbara-PA	4 males/3 females	ZUEC 24907; CFBH 46955; ZUEC 24908; CFBH 46958/CFBH 46750, 46782, 46784	252.147; 252.153; 252.156; 252.157/252.166; 252.190; 252.192
CZ Pep-L1B	Dom Eliseu-PA	2 males/-	CFBH 46703, 46711	92.383; 92.391
	São Pedro da Água Branca-PA	1 male/2 females	HUFMA 2295/MNRJ 24258; HUFMA 2290	92.332/92.4; 92.327
Lineage 1B	Riachão-MA ^1^	1 male/-	CFBH 46777	92.421
	Urbano Santos-MA	2 males/1 female	ZUEC 13362, 13363/ZUEC 13364	92.60; 92.61/92.62

^1^ Only the PepSat3-152 was mapped.

**Table 2 biomolecules-15-00876-t002:** Satellitome of *Physalaemus ephippifer*. List of all detected satDNAs and their main characteristics. The abundance of each element was measured using an input with sequences from both male and female genomes.

satDNA	Monomer Size (bp)	Abundance (%)	Average Divergence (%) Between Total Monomers	A+T Percentage *^2^	GenBank Accession Number
PepSat1-21	21	1.55	9.58	61.9	PV463924
PepSat2-97	97	0.96	7.25	55.67	PV463925
PepSat3-152 *	152	0.86	1.58	59.21	PV463926
PepSat4-149 *	149	0.71	9.03	61.07	PV463927
PepSat5-1310	1310	0.45	15.18	54.73	PV463928
PepSat6-1546	1546	0.40	10.25	52.78	PV463929
PepSat7-902	902	0.36	13.3	48	PV463930
PepSat8-36	36	0.28	10.88	52.78	PV463931
PepSat9-49	49	0.25	17.33	53.06	PV463932
PboSat8-92 ^1^-v-Pep	95	0.249	9.98	55.79	PV463933
PepSat11-237 *	237	0.242	9.16	40.93	PV463934
PepSat12-73	73	0.23	10.52	65.75	PV463935
PepSat13-49	49	0.219	7.9	55.10	PV463936
PepSat14-100	100	0.21	5.38	57	PV463937
PboSat05-35 ^1,2^-v-Pep	36	0.193	3.43	63.89	PV463938
PepSat16-147 *	147	0.192	7.9	45.58	PV463939
PepSat17-162	162	0.19	10.8	53.09	PV463940
PepSat18-1073	1073	0.177	4.26	52.1	PV463941
PepSat19-629	629	0.172	7.6	56.92	PV463942
PepSat20-22	22	0.16	9.83	54.55	PV463943
PepSat21-49	49	0.14	16.37	55.1	PV463944
PcP190 1a ^3,4^	190	0.13	2.08	52.11	PV463945
PboSat36-39 ^1^-v-Pep	39	0.129	9.31	56.41	PV463946
PepSat24-620 *	620	0.122	7.98	56.13	PV463947
PepSat25-282 *	282	0.11	4.61	45.39	PV463948
PepSat26-908	908	0.097	9.84	61.45	PV463949
PepSat27-280	280	0.092	2.11	62.5	PV463950
PboSat22-90 ^1^-v-Pep	31	0.08	5.45	58.06	PV463951
PboSat11-39 ^1,2^-v-Pep	38	0.076	3.6	52.63	PV463952
PepSat30-350 *	350	0.075	4.13	59.71	PV463953
PepSat31-709	709	0.057	6.04	59.1	PV463954
BBR86 v-Pep ^5^	41	0.05	8.18	56.1	PV463955
PepSat33-30	30	0.049	16.45	56.67	PV463956
PepSat34-57	57	0.048	17.57	45.61	PV463957
PepSat35-28	28	0.047	12.43	42.86	PV463958
PepSat36-739	739	0.04	5.38	61.3	PV463959
PepSat37-101	101	0.04	9.96	60.4	PV463960
PepSat38-39	39	0.039	5.63	41.03	PV463961
PepSat39-90	90	0.038	3.24	57.78	PV463962
PboSat12-91 ^1,2^-v-Pep	45	0.03	4.33	64.44	PV463963
PboSat17-93 ^1,2^-v-Pep	93	0.027	5.67	63.44	PV463964
PepSat42-41	41	0.026	9.94	46.34	PV463965
PepSat43-54	54	0.024	7.31	38.89	PV463966
PepSat44-16	16	0.023	12.42	37.5	PV463967
PepSat45-33	33	0.022	3.18	66.67	PV463968
PepSat46-47	47	0.02	9.78	53.19	PV463969
PepSat47-74	74	0.019	2.83	43.24	PV463970
PepSat48-27	27	0.019	4.02	48.15	PV463971
PepSat49-68	68	0.018	5.55	60.29	PV463972
PepSat50-32	32	0.017	5.19	65.63	PV463973
PepSat51-109	109	0.015	11.5	60.55	PV463974
PepSat52-30	30	0.014	5.19	56.67	PV463975
PepSat53-643	643	0.013	2.36	57.7	PV463976
PepSat54-31	31	0.012	10.08	48.39	PV463977
PepSat55-33	33	0.011	16.07	48.48	PV463978
PepSat56-31	31	0.0092	5.25	61.29	PV463979
PepSat57-158	158	0.009	8.29	60.76	PV463980
PepSat58-45	45	0.009	4.75	46.67	PV463981
PepSat59-30	30	0.009	5.56	36.67	PV463982
PepSat60-110	110	0.009	5.98	51.82	PV463983
PepSat61-23	23	0.008	4.93	43.48	PV463984
PepSat62-56	56	0.004	3.64	35.71	PV463985

* satDNAs mapped by FISH assays. *^2^ A+T content from the consensus sequence. ^1^ [52]; ^2^ [53]; ^3^ [25]; ^4^ [26]; ^5^ [54].

## Data Availability

The data used in this study are available in the GenBank Nucleotide Database at https://www.ncbi.nlm.nih.gov/genbank/, under accession numbers PV463924-PV463994 and in the Sequence Read Archive (SRA) under the accession numbers (SRR33457058, SRR33387195, and SRR33387196).

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
