# Peer review of "New Insights into the Sex Chromosome Evolution of the Common Barker Frog Species Complex (Anura, Leptodactylidae) Inferred from Its Satellite DNA Content"

_biomolecules, 2025, doi:10.3390/biom15060876_

Round 1
Reviewer 1 Report
Comments and Suggestions for Authors
The paper is correctly dimensioned, as is the use of modern cytogenomic and bioinformatics methodologies to obtain interesting and consistent results for the evolution of the sex chromosomes of Braker's frog species complex. The observation of nine satDNA families compatible with those described in taxonomically distant species stands out among the results, suggesting their prospecting in other anuran genera. I consider this work to be relevant to this group of vertebrates. I would like to ask the authors of Figures 1B and 4E, specimen SMRP 92.62 (L1B Ph cuvieri) whether the difference between the homologues of pair 8 is due to an artifact of the technique or to another factor?
Author Response
Dear Reviewer,
We are deeply grateful for the time and thoughtful comments dedicated to our manuscript. We believe that your suggestions will significantly enhance the quality of our work. Please find below our detailed responses to your comments.
Comment: The paper is correctly dimensioned, as is the use of modern cytogenomic and bioinformatics methodologies to obtain interesting and consistent results for the evolution of the sex chromosomes of Braker's frog species complex. The observation of nine satDNA families compatible with those described in taxonomically distant species stands out among the results, suggesting their prospecting in other anuran genera. I consider this work to be relevant to this group of vertebrates. I would like to ask the authors of Figures 1B and 4E, specimen SMRP 92.62 (L1B Ph cuvieri) whether the difference between the homologues of pair 8 is due to an artifact of the technique or to another factor?
Answer: Actually, this is not an artifact but a real difference in NOR size. The number, size, and position of NORs are highly variable in some of the lineages—particularly in lineage 1 of “Ph. cuvieri”, as described by Quinderé et al. (2009). In the specimen SMRP 92.62, one homologue of pair 8 has a larger NOR, as shown in the inset of Figure 4. Accordingly, chromosome pair 8 is heteromorphic with respect to the size of the secondary constriction (as seen in Figure 1B) and the size of PepSat25 signals (as seen in Figure 4E), which colocalize with the NORs.

Reviewer 2 Report
Comments and Suggestions for Authors
The manuscript deals with the sex chromosome evolution of three lineages of the Physalaemus cuvieri – Physalaemus ephippifer complex be means of different satellite DNAs.
I consider the manuscript to be relevant and all sections are clearly organized. It presents an interesting hypothesis of chromosomal evolution in Physalaemus ephippifer.
A few comments follow below:
Line 323-324: “Firstly, in Ph. ephippifer, the PepSat16-147 probe revealed signals that co-localized with all NORs on chromosomes 7 and 8 and on the Z and W chromosomes (Figure 4A).”
The signal is not seen on pair 7 and pair 8 corresponds to the sexual pair.
Lines 324-326: “Notably, the signals characteristic of the sex chromosomes were stronger than those found on chromosomes 7 and 8 (Figure 4C).”
Cross out “and 8”. Change “(Figure 4C)” by (Figure 4A).”
Lines 326-327: “Additionally, in this lineage, a PepSat16-147 cluster not associated to a NOR was found on the interstitial region of the Z chromosome.”
It could be added to the legend of the figure that is indicated with an arrow.
In Figure 4C, is the signal seen on one of the chromosomes in pair 3 a technical artifact?
If so, could this be clarified in the figure?
Lines 596-599: This finding suggests the possibility of a translocation event, where a segment composed of a NOR and the terminal PepSat24-620 region was transferred from Wq to Wp, a rearrangement that may explain the most prominent difference between the sex chromosomes of Ph. ephippifer (Figure 7).
Could these findings also be explained by a pericentric inversion with another terminal break in the short arm?
Author Response
Dear Reviewer,
We are deeply grateful for the time and thoughtful comments dedicated to our manuscript. We believe that your suggestions will significantly enhance the quality of our work. Please find below our detailed responses to your comments.
Comment: The manuscript deals with the sex chromosome evolution of three lineages of the Physalaemus cuvieri – Physalaemus ephippifer complex be means of different satellite DNAs. I consider the manuscript to be relevant and all sections are clearly organized. It presents an interesting hypothesis of chromosomal evolution in Physalaemus ephippifer.
A few comments follow below:
1) Line 323-324: “Firstly, in Ph. ephippifer, the PepSat16-147 probe revealed signals that co-localized with all NORs on chromosomes 7 and 8 and on the Z and W chromosomes (Figure 4A).” The signal is not seen on pair 7 and pair 8 corresponds to the sexual pair.
Answer: This error was corrected in the revised manuscript. This statement should refer to CZ Pep-L1B and not P. ephippifer.
Comment: 2) Lines 324-326: “Notably, the signals characteristic of the sex chromosomes were stronger than those found on chromosomes 7 and 8 (Figure 4C).” Cross out “and 8”. Change “(Figure 4C)” by (Figure 4A).”
Answer: We have revised the structure of the paragraph in the updated version of the manuscript.
Comment: 3) Lines 326-327: “Additionally, in this lineage, a PepSat16-147 cluster not associated to a NOR was found on the interstitial region of the Z chromosome.” It could be added to the legend of the figure that is indicated with an arrow.
Answer: In the revised version of the manuscript, this information has been incorporated into the figure legend.
Comment: 4) In Figure 4C, is the signal seen on one of the chromosomes in pair 3 a technical artifact? If so, could this be clarified in the figure?
Answer: Yes, the signal resulted from an artifact; we have clarified this point in the figure legend in the revised version of the manuscript.
Comment: 5) Lines 596-599: This finding suggests the possibility of a translocation event, where a segment composed of a NOR and the terminal PepSat24-620 region was transferred from Wq to Wp, a rearrangement that may explain the most prominent difference between the sex chromosomes of Ph. ephippifer (Figure 7). Could these findings also be explained by a pericentric inversion with another terminal break in the short arm?
Answer: Additional markers on Zp and Wp are still lacking, which prevents a proper analysis, but the markers currently available for Zq and Wq are indeed consistent with the hypothesis of pericentric inversion. In the revised version of the manuscript, we have rephrased this paragraph to incorporate this possibility.

Reviewer 3 Report
Comments and Suggestions for Authors
The study by Souza et al. investigates satellitome of a frog species complex and, as a unique organisms, provides a valuable insight into the dynamics and evolution of satDNA. The authors performed an in-depth analysis of the total satDNA content using bioinformatic approaches and validated this with extensive FISH experiments. The background and aim of the study are clearly described. The methods and results are also well presented, with several remarks that should be addressed. It is not entirely clear how the authors specifically selected the satDNA for the FISH experiments, this should be emphasized more. It would also be interesting to include the two most abundant satDNAs. Mainly because the authors are referring to the potential centromeric function of the third most abundant satDNA, and often centromeric satDNA are the ones that are most abundant in the genome. One suggestion would also be to shorten the lengthy discussion in several parts, except for the section on the link between satDNA and sex chromosomes, which is very interesting and well written. Overall, I think these results will be of interest to many readers.
Major remarks:
1. Please provide information on species genome sizes and the potential availability of genome assemblies, as I see that there are none yet. This could perhaps be included in the introduction.
2. The authors should comment if they tried different coverages of RE2 analysis and why did they decide on 6 million reads as optimal? This is important as several studies have found differences in the amount of satDNA found when different coverages were used.
3. In all the karyograms presented, how could the authors be sure which chromosome pair is which and what its equivalent is in the different species (except ZW chr)? Maybe add this in the methods section.
4. How do the authors comment on the presence of polymorphic satDNA clusters in some species? Is this a feature of frogs and these genomes? It would be interesting to discuss this part in more detail.
5. Lines 435-447: Consider shortening this part of the discussion as these challenges in the satDNA study are not the focus of this paper and there are already many studies using similar approaches.
6. Lines 454-474: Please shorten the part on nomenclature. Recent studies largely use the proposed nomenclature system, and I do not think it is of primary importance to raise this issue.
7. Part of the discussion on centromeric satDNA should comment on the two most abundant satDNAs and be better linked to the other similar examples of centromeric satDNAs in different species.
8. It would be nice to include other studies that infer species evolution from satellitome analysis and compare them with your results.
Minor remarks:
- There are several specimens listed in Table 1. Perhaps indicate what they were used for (chromosome preparations?) if only one male and one female were sequenced.
- Add the accession numbers of the deposited reads.
- Line 192: "divergent primers", do you mean satDNA-specific primers?
- FISH method (lines 202-204): perhaps add a sentence describing from which tissue and how the slides were prepared, fixed, etc.
- Ag-NOR method (Line 209): It might be beneficial to add some specific details of how it was done. I am wondering if you noticed any interference with the FISH signals on these slides when combined with the Ag-NOR method?
- Lines 233-236: Which is the female satDNA with the highest abundance (PepSat30 or PepSat16)? This part is a bit confusing as both are mentioned in the same context.
- Line 246: "with only the PcP190 monomer exceeding 100 bp" - What does this 100bp refer to, the length variation or the length of the satDNA monomers? Consider a more precise specification to provide clarity.
- Line 256: Change "prospected" to detected/characterised/found
- Line 260-261: Repeated from the Table title, consider deleting in one place.
- Line 267: consider rephrasing to "...on primary constiction of the centromeric region of all chromosomes."
- Lines 323-324: The information in this sentence is only visible in Ph. ephippifer and Figure 4A, as marked, or in the entire Figure 4?
- Figure 4B is not referenced, but is explained after Figure 4C.
- Lines 326-327: Where is this additional cluster visible in the interstitial region in Figure 4C?
- In the caption of Figure 4A, add the description of what the full arrowhead represents (as is written for empty arrow in Figure 4B).
- The same comment applies to Figure 6, which also lacks the description of the full arrowhead.
- Add in the discussion the potential mechanistic implications of the NOR regions, especially since they seem to be very relevant features of karyotypic diversity.
Author Response
Dear Reviewer,
We are deeply grateful for the time and thoughtful comments dedicated to our manuscript. We believe that your suggestions will significantly enhance the quality of our work. Please find below our detailed responses to your comments.
Comment: The study by Souza et al. investigates satellitome of a frog species complex and, as a unique organisms, provides a valuable insight into the dynamics and evolution of satDNA. The authors performed an in-depth analysis of the total satDNA content using bioinformatic approaches and validated this with extensive FISH experiments. The background and aim of the study are clearly described. The methods and results are also well presented, with several remarks that should be addressed. It is not entirely clear how the authors specifically selected the satDNA for the FISH experiments, this should be emphasized more. It would also be interesting to include the two most abundant satDNAs. Mainly because the authors are referring to the potential centromeric function of the third most abundant satDNA, and often centromeric satDNA are the ones that are most abundant in the genome. One suggestion would also be to shorten the lengthy discussion in several parts, except for the section on the link between satDNA and sex chromosomes, which is very interesting and well written. Overall, I think these results will be of interest to many readers.
Major remarks:
1) Please provide information on species genome sizes and the potential availability of genome assemblies, as I see that there are none yet. This could perhaps be included in the introduction.
Answer: In response to the reviewer’s suggestion, we clarify that the genome size of Ph. ephippifer had not been previously studied. We therefore estimated it using kmers with GenomeScope2, and have added this information to the Material and Methods, Results, and Discussion sections.
Regarding the availability of genome assemblies, we attempted to assemble the genomes of both male and female samples using SOAPdenovo2. However, due to the nature of the Illumina libraries and their relatively low coverage (10-20x based on k-mer counts; see GenomeScope2 results in the supplementary figure S1), the resulting assemblies were highly fragmented, limiting their utility for subsequent analyses.
We present some statistics from the male assembly that highlight its high fragmentation.
Main genome scaffold total: 4715011
Main genome scaffold sequence total: 2645.076 MB
Main genome scaffold L50/N50: 377520/1.714 KB
Number of scaffolds > 50 KB: 2
Minimum Scaffold length Number of Scaffold Total Scaffold Length (bp)
< 250 bases 4,715,011 2,645,075,730
> 250 bases 1,581,894 2,209,335,781
> 500 bases 1,148,713 2,051,739,506
> 5 Kb 57,853 438,728,219
> 10 Kb 8,277 111,360,841
> 25 Kb 169 5,066,875
> 50 Kb 2 128,042
Comment: 2) The authors should comment if they tried different coverages of RE2 analysis and why did they decide on 6 million reads as optimal? This is important as several studies have found differences in the amount of satDNA found when different coverages were used.
Answer: We performed several analyses (approximately ten, considering those based solely on female data, male data, or both), and in our case, RepeatExplorer (RE) was consistently able to process fewer than 3,000,000 random reads from our samples. Since RE had more reads available than it could analyze, we believe this factor did not influence our results, particularly regarding the amount of satDNA recovered. We have included a comment on this in section 2.3 of the revised manuscript.
Comment: 3) In all the karyograms presented, how could the authors be sure which chromosome pair is which and what its equivalent is in the different species (except ZW chr)? Maybe add this in the methods section.
Answer: Based on the combination of chromosome size, morphology, and previously obtained cytogenetic markers (e.g., C-bands, NORs, PcP190 satDNA, PepBS repetitive DNA, and a probe contructed from microdissected Z chromosomes of Ph. ephippifer), we were able to distinguish most chromosome pairs, including the sex chromosomes of Ph. ephippifer and their homeologous chromosome 9 of L1B. Notably, the inference of homeology between these chromosomes was previously proposed by Gatto et al. (2021), and arose from the chromosome mapping of a probe (Zqper) constructed from microdissected Z chromosomes of Ph. ephippifer. Interestingly, in our present study we could observe that the same regions mapped by the Zqper probe were also revealed by the PepSat11 probe. We have added a note at the end of the Materials and Methods section to clarify this point.
Comment: 4) How do the authors comment on the presence of polymorphic satDNA clusters in some species? Is this a feature of frogs and these genomes? It would be interesting to discuss this part in more detail.
Answer: We believe the Reviewer is referring to the satDNAs associated with NORs. This is indeed an interesting point, as variation in NOR size and number has been widely reported for anurans. The satDNAs we found associated with NORs consistently corresponded with the variation detected by the Ag-NOR method (which reveals nucleolar protein associated with NORs). We have included a brief commentary on this in the revised manuscript – section 4.4.2.
Comment: 5) Lines 435-447: Consider shortening this part of the discussion as these challenges in the satDNA study are not the focus of this paper and there are already many studies using similar approaches.
Answer: In the revised version, we have shortened this section of the manuscript.
Comment: 6) Lines 454-474: Please shorten the part on nomenclature. Recent studies largely use the proposed nomenclature system, and I do not think it is of primary importance to raise this issue.
Answer: We acknowledge that this issue was discussed with more emphasis than initially expected. However, we chose to include it because we encountered certain challenges during our analyses that highlighted the importance of methodological caution. Although numerous satellitomes have been described in recent years, we believe that if well-established practices are not applied carefully, future analyses may become increasingly problematic.
As an illustrative example, we refer to Supplementary Figure S5, which highlights inconsistencies in the satellitome data of Proceratophrys boiei. Two studies have been published on the satellitome of this species [50,51]: the first describes 28 satellite DNAs (satDNAs), while the second reports 226. We suspect that most, if not all, of the satDNAs described in both studies are duplicates, as abundance—one of the criteria used for naming—can fluctuate among individuals from different localities. Furthermore, as shown in the supplementary figure, study [51] describes the same satDNA as two distinct elements, when in fact they are reverse complements of each other. These inconsistencies were identified primarily due to similarities with satDNAs found in Ph. ephippifer, suggesting that the actual number of errors may be even higher.
We believe this cautionary note is important, as such inconsistencies may hinder future studies, particularly those aiming to investigate conserved satDNAs across species that diverged long ago.
Comment: 7) Part of the discussion on centromeric satDNA should comment on the two most abundant satDNAs and be better linked to the other similar examples of centromeric satDNAs in different species.
Answer: We understand that, in most cases, the centromeric satDNA is recovered as the most abundant in RepeatExplorer analysis; however, this is not a general rule. We cannot rule out the possibility that PepSat1 and PepSat2 are also located in centromeric or pericentromeric regions. In fact, primers were designed for PepSat2, but we were unable to amplify it efficiently to generate a probe, and thus decided to proceed with the manuscript in its absence. Furthermore, the literature includes several examples in which the most abundant satDNA is either not located in the centromere of all chromosomes or not restricted to centromeric regions. For instance, Bardella et al. (2020) reported that HhiSat1-184 was mapped to the pericentromeric region of some chromosomes, was absent in others, and also showed signals in interstitial and telomeric regions. Another example is the study by Alex-Mata et al. (2025), in which the three most abundant satDNAs exhibited centromeric localization in Microtus thomasi, although none of them were present in all chromosomes, unlike some less abundant sequences. Interestingly, in Pontastacus leptodactylus, PlSat3-411 was the only satDNA mapped to the centro-/pericentromeric region of all chromosomes in the karyotype. In contrast, PlSat1-21—despite having the same monomer length (21 bp) and being about eight times more abundant (10.91%) than PlSat3-411 (1.29%)—was also mapped to centromeric regions but not across all chromosomes (Boštjančić et al., 2021). Lastly, the differences we observed among the most abundant satDNAs were not particularly pronounced, and depending on the lineage analyzed, this ranking may vary, as suggested by ongoing studies currently being conducted in our laboratory. We have added some comments on this issue in the revised manuscript – section 4.4.1.
Comment: 8) It would be nice to include other studies that infer species evolution from satellitome analysis and compare them with your results.
Answer: We have included a comment in section 4.2 to highlight the relevance of satDNAs in interspecific comparisons and their usefulness in discussions of karyotype evolution. However, we emphasize here that we addressed the phylogenetic relationships within the species complex Ph. cuvieri – Ph. ephippifer using mitochondrial DNA sequences and RAD markers (Nascimento et al., 2019; Souza et al., 2024), which provided robust evolutionary inferences that guide our interpretation of chromosomal evolution in this group.
Comment: Minor remarks:
9) There are several specimens listed in Table 1. Perhaps indicate what they were used for (chromosome preparations?) if only one male and one female were sequenced.
Answer: In the first sentence of the Materials and Methods section, where Table 1 is cited, we initially stated that the specimens were used for FISH experiments. To clarify this point, we have rephrased the sentence accordingly in the revised manuscript.
Comment: 10) Add the accession numbers of the deposited reads.
Answer: We provided this information in the new version.
Comment: 11) Line 192: "divergent primers", do you mean satDNA-specific primers?
Answer: The intention was to indicate that the forward and reverse primers are oriented in opposite directions based on the monomer sequence, which is one of the approaches used to corroborate their tandem organization. We have revised the text to clarify this point.
Comment: 12) FISH method (lines 202-204): perhaps add a sentence describing from which tissue and how the slides were prepared, fixed, etc.
Answer: In the revised version of the manuscript, we have included additional details regarding the preparation of the sample.
Comment: 13) Ag-NOR method (Line 209): It might be beneficial to add some specific details of how it was done. I am wondering if you noticed any interference with the FISH signals on these slides when combined with the Ag-NOR method?
Answer: In the revised version of the manuscript, we expanded the description of the Ag-NOR method to provide a more detailed explanation of the procedure. Based on the results obtained from slides subjected either exclusively to the Ag-NOR method or to both FISH and the Ag-NOR method, we have noted that both approaches revealed the same NOR pattern. Therefore, FISH does not interfere with silver staining of the NORs, and the sequential application of both techniques ensure accurate identification of the chromosomes of interest.
Comment: 14) Lines 233-236: Which is the female satDNA with the highest abundance (PepSat30 or PepSat16)? This part is a bit confusing as both are mentioned in the same context.
Answer: In fact, PepSat30-350 exhibits the highest female-to-male abundance ratio (0.11% in females vs. 0.04% in males), as shown in Table S2. The mention of PepSat16-147 refers to the fact that, among the seven satDNAs with a female-biased abundance pattern (i.e., PepSat16, PepSat20, PepSat25, PepSat30, PepSat45, PepSat53, and PepSat57), PepSat16 had the highest overall genomic abundance. To make it clearer, we have included the overall abundance values for each of these satDNAs in the corresponding paragraph.
Comment: 15) Line 246: "with only the PcP190 monomer exceeding 100 bp" - What does this 100bp refer to, the length variation or the length of the satDNA monomers? Consider a more precise specification to provide clarity.
Answer: It referred to the monomer length. We revised the paragraph to make it clearer.
Comment: 16) Line 256: Change "prospected" to detected/characterised/found.
Answer: Done.
Comment: 17) Line 260-261: Repeated from the Table title, consider deleting in one place.
Answer: Corrected in the revised version.
Comment: 18) Line 267: consider rephrasing to "...on primary constiction of the centromeric region of all chromosomes."
Answer: Done.
Comment: 19) Lines 323-324: The information in this sentence is only visible in Ph. ephippifer and Figure 4A, as marked, or in the entire Figure 4?
Answer: This section had a minor error, which we have corrected in the revised version. In Ph. ephippifer (Figure 4A), the PepSat16 probe signal was observed only on the Z and W chromosomes. In contrast, PepSat16 signals were observed in 3 chromosome pairs of the CZ Pep-L1B specimens (chromosomes 7, 8, and ZW), which were mistakenly referred to as Ph. ephippifer in the original manuscript.
Comment: 20) Figure 4B is not referenced, but is explained after Figure 4C.
Answer: In the revised version, we have referenced Figure 4B and reorganized the paragraph to improve clarity.
Comment: 21) Lines 326-327: Where is this additional cluster visible in the interstitial region in Figure 4C?
Answer: In fact, that paragraph was somewhat unclear and included a minor error regarding the figure reference and the lineage name. In the revised version, we reorganized the paragraph to correct these issues. Additionally, in the figure, this signal is indicated in Ph. ephippifer by a filled arrowhead, which was not mentioned in the original legend; we have included this information in the revised version.
Comment: 22) In the caption of Figure 4A, add the description of what the full arrowhead represents (as is written for empty arrow in Figure 4B).
Answer: It was included in the revised version.
Comment: 23) The same comment applies to Figure 6, which also lacks the description of the full arrowhead.
Answer: It was included in the revised version.
Comment: 24) Add in the discussion the potential mechanistic implications of the NOR regions, especially since they seem to be very relevant features of karyotypic diversity.
Answer: In the revised version of the manuscript, we expanded the paragraph in the Discussion section that addresses this issue (lines 720-738). However, as further analyses, including additional techniques and other lineages (with different levels of NOR variation), are still necessary to properly address the mechanisms and evolutionary implications of these NORs, we were careful to avoid speculation.
References
Alex-Mata, G.; Montiel, E.E.; Mora, P.; Yurchenko, A.; Rico-Porras, J.M.; Anguita, F.; Palomo, F.; MArchal, J.A.; Rovatsos, M.; Sánchez, A. 2025. Satellitome analysis on Microtus thomasi (Arvicolinae) genome, a mammal species with high karyotype and sex chromosome variations. Genome. 68:1-13.
Bardella, V.B.; Milani, D.; Cabral-de-Mello, D.C. 2020. Analysis of Holhymenia histrio genome provides insight into the satDNA evolution in an insect with holocentric chromosomes. Chromosome Res. 28:369-380.
Boštjančić, L.L.; Bonassin, L. Anušić, L.; Lovrenčić, L.; Besendorfer, V.; Maguire, I.; Grandjean, F.; Austin, C.M.; Greve, C.; Hamadou, A.B.; Mlinarec, J. 2021. The Pontastacus leptodactylus (Astacidae) repeatome provides insight into genome evolution and reveals remarkable diversity of satellite DNA. Front. Genet. 11:611745.
Gatto, K.P.; Souza, L.H.B.; Nascimento, J.; Suárez, P.; Lourenço, L.B. 2021. Comparative Mapping of a New Repetitive DNA Sequence and Chromosome Region-Specific Probes Unveiling Rearrangements in an Amazonian Frog Complex. Genome 64: 857–868.
Nascimento, J.; Lima, J.D.; Suárez, P.; Baldo, D.; Andrade, G.V.; Pierson, T.W.; Fitzpatrick, B.M.; Haddad, C.F.B.; Recco-Pimentel, S.M.; Lourenço, L.B. 2019. Extensive Cryptic Diversity within the Physalaemus Cuvieri–Physalaemus Ephippifer Species Complex (Amphibia, Anura) Revealed by Cytogenetic, Mitochondrial, and Genomic Markers. Front. Genet. 10: 719.
Souza, L.H.B.; Pierson, T.W.; Tenório, R.O.; Ferro, J.M.; Gatto, K.P.; Silva, B.C.; De Andrade, G.V.; Suárez, P.; Haddad, C.F.B.; Lourenço, L.B. 2024. Multiple Contact Zones and Karyotypic Evolution in a Neotropical Frog Species Complex. Sci. Rep. 14: 1119.

Reviewer 4 Report
Comments and Suggestions for Authors
The article by Souza and co-authors focuses on the evolutionary dynamics of chromosomes, particularly sex chromosomes, in three genetic lineages of anuran amphibians, which include two evolutionarily diverged lines and a lineage that arose from secondary contact between the first two. They conduct Whole Genome Sequencing for one of the species—Ph. ephippifer. Based on the obtained reads, the authors perform a bioinformatic analysis of the satelliteome for this species. Some satellites of particular interest hybridize in situ on metaphase chromosomes from three genetic lines across different geographical locations. The comparative analysis allows for the reconstruction of an evolutionary scenario regarding the evolution of sex chromosomes within the group and yields additional important results; for instance, identifying a satellite that is presumably associated with centromere function.
I found the article to be well-written and logically structured; it is illustrated with high-quality cytological photographs.
I have some comments regarding the section of the article related to identifying the supercluster formed by rDNA and satellite DNAs. The authors state “clusters in this supercluster showed similarity to nucleolar 40S rDNA.” It is likely that they meant to refer to 45S rDNA instead.
The authors also write, “the supercluster analysis generated by RepeatExplorer, which included 40S rDNA, suggests that these three types of repetitive DNA are part of the intergenic spacer (IGS) of nucleolar rDNA in Ph. ephippifer.” This very interesting result aligns well with cited works regarding the presence of satellite DNA in IGS in some organisms. It would be beneficial if the authors could elaborate on this result further. Currently, the article presents a diagram of the supercluster (Fig. 6), consisting of a cluster containing rRNA genes but excluding satellites (upper cluster), and several clusters including satellites and then again clusters containing rRNA genes without satellites. According to the last diagram in Fig. 7 there is an alternation involving PepSat16; it can be assumed that this is what occurs in IGS. In this case, it remains unclear how to reconcile the text with diagrams in Fig. 6 and Fig. 7. I would like to see a more detailed explanation from the authors about how the chromosome segment carrying the supercluster is structured and what constitutes a repeat unit within it? How do components of the supercluster occupy a significant portion of the chromosome according to FISH? What does a repeat unit of rRNA genes with satellites inside spacers look like?
Fig. 7 is difficult to interpret as it lacks clear labeling regarding which genetic line is being discussed; perhaps enhancing its clarity would be beneficial since it is key to this work. There are areas on Fif 7 where labels are very small; the same – very small are labels within circles for elements of the supercluster in Fig. 6.
On Fig. 6 “Pepsat” should be “PepSat”
A technical issue arose as I was unable to access supplementary materials for this article; they were not provided as a separate file with the manuscript and could not be opened via the link provided at the end of the article. Therefore, I lean towards recommending a re-review of this work due to my inability to access all materials for technical reasons.
Overall my opinion is that this is a well-written article that presents undeniable interest and deserves publication in Biomolecules after minor revisions.
Author Response
Dear Reviewer,
We are deeply grateful for the time and thoughtful comments dedicated to our manuscript. We believe that your suggestions will significantly enhance the quality of our work. Please find below our detailed responses to your comments.
Comment: The article by Souza and co-authors focuses on the evolutionary dynamics of chromosomes, particularly sex chromosomes, in three genetic lineages of anuran amphibians, which include two evolutionarily diverged lines and a lineage that arose from secondary contact between the first two. They conduct Whole Genome Sequencing for one of the species—Ph. ephippifer. Based on the obtained reads, the authors perform a bioinformatic analysis of the satelliteome for this species. Some satellites of particular interest hybridize in situ on metaphase chromosomes from three genetic lines across different geographical locations. The comparative analysis allows for the reconstruction of an evolutionary scenario regarding the evolution of sex chromosomes within the group and yields additional important results; for instance, identifying a satellite that is presumably associated with centromere function. I found the article to be well-written and logically structured; it is illustrated with high-quality cytological photographs.
1) I have some comments regarding the section of the article related to identifying the supercluster formed by rDNA and satellite DNAs. The authors state “clusters in this supercluster showed similarity to nucleolar 40S rDNA.” It is likely that they meant to refer to 45S rDNA instead.
Answer: Since the pioneer study of Rogers (1968), who determined that the rRNA precursor in amphibian species of Triturus and Ambystoma is a 40S RNA, this has been assumed to be representative of the precursor rRNA of amphibians in general (e.g., Wellauer et al., 1974; Dawid & Wellauer, 1976; Labhart & Reeder, 1990; Ajuh et al. 1991; Roger et al., 2002; Roger et al., 2003). For this reason, we also adopted this nomenclature, although it may not necessarily reflect the condition in all anuran species.
Comment: 2) The authors also write, “the supercluster analysis generated by RepeatExplorer, which included 40S rDNA, suggests that these three types of repetitive DNA are part of the intergenic spacer (IGS) of nucleolar rDNA in Ph. ephippifer.” This very interesting result aligns well with cited works regarding the presence of satellite DNA in IGS in some organisms. It would be beneficial if the authors could elaborate on this result further.
Answer: In the revised version of the manuscript, we provided additional commentary on this point. However, further studies are still needed to enable proper inferences, and we have suggested possible directions (which are indeed part of our ongoing research).
Comment: 3) Currently, the article presents a diagram of the supercluster (Fig. 6), consisting of a cluster containing rRNA genes but excluding satellites (upper cluster), and several clusters including satellites and then again clusters containing rRNA genes without satellites. According to the last diagram in Fig. 7 there is an alternation involving PepSat16; it can be assumed that this is what occurs in IGS. In this case, it remains unclear how to reconcile the text with diagrams in Fig. 6 and Fig. 7. I would like to see a more detailed explanation from the authors about how the chromosome segment carrying the supercluster is structured and what constitutes a repeat unit within it? How do components of the supercluster occupy a significant portion of the chromosome according to FISH? What does a repeat unit of rRNA genes with satellites inside spacers look like?
Answer: Figure 6, which presents the supercluster, represents a possible organization of the nucleolar rRNA genes and the IGS in Ph. ephippifer. However, it is important to highlight a few points: (1) the RepeatExplorer pipeline generates superclusters by assembling reads from all different NOR sites, which may result in chimeric assemblies in cases where there is substantial variation among loci. That is precisely why it is so important to use FISH to map the sequences inferred by RepeatExplorer to compose the IGS. The chromosome mapping of the putative IGS elements to chromosomes of Ph. ephippifer did not show any variation among the three NOR sites (two on the W chromosome and one on the Z chromosome), suggesting they are relatively similar in molecular composition. However, our current dataset does not include fiber-FISH or other analyses that could provide sufficient resolution to determine the detailed organization of these sequences. In fact, we are currently conducting a study aimed at further exploring this question. (2) The reduction in the copy number of PepSat16 at the NOR located on the short arm of the W chromosome in the CZ Pep-L1B group, which is depicted in Figure 7, was inferred from FISH assays. The supercluster shown in Figure 6 was assembled using samples from Ph. ephippifer, not from CZ Pep-L1B. Thus, it is reasonable to assume that the molecular composition of the NORs of CZ Pep-L1B are not homogeneous, with the NOR on Wp differing from the other. At last, we should emphasize that FISH using rDNA probes are already available for Ph. ephippifer and L1B in our previous studies (Gatto et al., 2021), as we discuss in the section 4.4.2. To improve the clarity of Figure 7, we increased the font size and added labels indicating where the FISH signal was absent.
Comment: 4) Fig. 7 is difficult to interpret as it lacks clear labeling regarding which genetic line is being discussed; perhaps enhancing its clarity would be beneficial since it is key to this work. There are areas on Fif 7 where labels are very small; the same – very small are labels within circles for elements of the supercluster in Fig. 6.
Answer: We revised Figures 6 and 7 to improve the clarity and visualization of the information presented.
Comment: 5) On Fig. 6 “Pepsat” should be “PepSat”.
Answer: Corrected in the revised version.
Comment: A technical issue arose as I was unable to access supplementary materials for this article; they were not provided as a separate file with the manuscript and could not be opened via the link provided at the end of the article. Therefore, I lean towards recommending a re-review of this work due to my inability to access all materials for technical reasons.
Overall my opinion is that this is a well-written article that presents undeniable interest and deserves publication in Biomolecules after minor revisions.
Answer: We are sorry for that. We have submitted the supplementary material as a PDF file in the Supplementary Files Section. To make access easier, we also have also included the supplementary file in the response letter we attached here.
References
Ajuh, P.M., Heeney, P.A., Maden, B.E. 1991. Xenopus borealis and Xenopus laevis 28S ribosomal DNA and the complete 40S ribosomal precursor RNA coding units of both species. Proc Biol Sci., 245(1312): 65-71. doi: 10.1098/rspb.1991.0089.
Dawid,I.B., Wellauer, P.K. 1976. A re-investigation of 5′å3′ polarity in 40S ribosomal RNA precursor of Xenopus laevis Cell, 8: 443-448.
Labhart, P., Reeder, R.H. 1990. A point mutation uncouples RNA 3'-end formation and termination during ribosomal gene transcription in Xenopus laevis. Genes Dev., 4(2)): 269-76. doi: 10.1101/gad.4.2.269. PMID: 1692557.
Roger, B., Moisand, A., Amalric, F. et al. 2003. Nucleolin provides a link between RNA polymerase I transcription and pre-ribosome assembly. Chromosoma 111: 399–407 (2003). https://doi.org/10.1007/s00412-002-0221-5
Rogers, M.E. 1968. Ribonucleoprotein particles in the amphibian oocyte nucleus: possible intermediates in ribosome synthesis. The Journal of Cell Biology, 36: 421–432. doi:10.1083/jcb.36.3.421
Roger, B., Moisand, A., Amalric, F., Bouvet, P. 2002. rDNA transcription during Xenopus laevis oogenesis. Biochem. Biophys. Res. Commun. 290(4): 1151-60. doi: 10.1006/bbrc.2001.6304. PMID: 11811983.
Wellauer, P.K., Dawid, I.B., Kelley, D.E., Perry, R.P. 1974. Secondary structure maps of ribosomal RNA. II. Processing of mouse L-cell ribosomal RNA and variations in the processing pathway. J Mol Biol., 89(2):397-407. doi: 10.1016/0022-2836(74)90527-0. PMID: 4475117.

Round 2
Reviewer 3 Report
Comments and Suggestions for Authors
The authors have responded fully to my comments and have made all the necessary changes to the manuscript.
I have found only a few textual errors:
Lines 247-249: end of sentence missing; add something like "...was possible."
Line 378: double punctuation
Line 387: missing comma and lowercase after - "Conversely, signals..."
Line 603: additional space before the punctuation
Line 691: missing space between two words "...in which a..."
Lines 740-741: check the end of the sentence and the punctuation. It also appears that part of the sentence after "...chromosomes of Ph." is missing
Author Response
Dear Reviewer,
Thank you for the careful revision and helpful corrections. We made all the requested changes.
Lines 247-249: end of sentence missing; add something like "...was possible."
Answer: Done.
Line 378: double punctuation
Answer: Corrected.
Line 387: missing comma and lowercase after - "Conversely, signals..."
Answer: We rephrased this paragraph to make it clearer.
Line 603: additional space before the punctuation
Answer: Corrected.
Line 691: missing space between two words "...in which a..."
Answer: Corrected.
Lines 740-741: check the end of the sentence and the punctuation. It also appears that part of the sentence after "...chromosomes of Ph." is missing
Answer: We rephrased this sentence.